# Beyond the Bases: Unleashing Overlapping DNA Tokenization via Unified Linear-Time Autoregressive

## Abstract

The domain of genomic language models (gLMs) has advanced rapidly, with models pretrained on diverse multi-species genomic corpus demonstrating their remarkable capabilities. While the effect on long-context modeling of simplest nucleotide-level tokenization has already been proven, overlapping k-mer, which provides richer neighborhood information, has been neglected in the existing gLM designs. Herein, we provide a thorough revisit of the overlapping tokenization and present HGDNA, a hybrid linear attention gLM under a unified causal language modeling (CLM) paradigm across pretraining and fine-tuning through species classification auxiliary task and shared class tokens. HGDNA provides superior capability across various classification, zero-shot embedding, and instruction-based sequence design tasks, demonstrating its robust performance and notable efficiency across both short-range and long-range tasks.

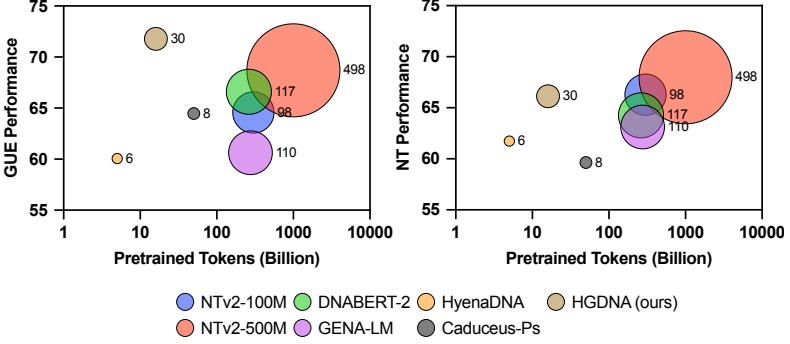

Figure 1: Performance of different gLMs across GUE benchmark (28 tasks) and revised Nucleotide Transformer benchmark (18 tasks), larger point size denotes larger parameter scale, with exact values annotated on their right side by Million (M). The proposed HGDNA with 30M parameters provides competitive or superior performance compared to other baselines

## 1 Introduction

Recent achievements in natural language processing (NLP) with the rapid development of large language models (LLMs) reveals the potential of extensive self-supervised training with massive volume of corpus and fine-grained domain-specific supervised fine-tuning (SFT) data (Devlin et al., 2019; OpenAI et al., 2024; Grattafiori et al., 2024; Qwen et al., 2025; DeepSeek-AI et al., 2025; Dong et al., 2024; Li et al., 2025). Following the achievements made by LLMs, broader domains have made noticeable strides in domain-specific LLM or multi-modality LLM (MLLM), e.g., computer vision (CV) (Yin et al., 2024), recommendation system (Zhang et al., 2024), and clinical question answering (QA) (Singhal et al., 2025). Since biological sequences are naturally compatible with NLP technologies, there are already several remarkable achievements have been made in both amino acid sequences (Lin et al., 2023), RNA sequences (Chen et al., 2022), and DNA sequences (Wu et al., 2025; Brixi et al., 2025). Thanks to the large amount of sequencing data, the

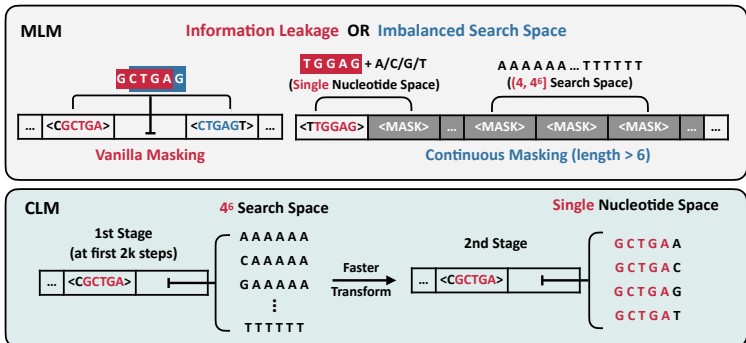

Figure 2: The overview of overlapping 6-mer pretraining in MLM and CLM objective. Due to the overlapping encoding scheme, vanilla **MLM** masking introduces critical information leakage, while contiguous masking can inevitably lead to an imbalanced masking recovery difficulty. In contrast, **CLM** pretraining prevents bidirectional information leakage and provides a smoother transition between overlapping pattern learning (stage 1) and single nucleotide modeling (stage 2).

recent genomic foundation model can annotate and understand various regulators, functional contigs, non-coding regions, as well as the long-range interactions between them, revolutionizing the pipeline of genomic analysis.

Under the context of genomics language modeling, a common and well-established strategy is to adapt methodologies from broader NLP domains, e.g., DNABERT-2 (Zhou et al., 2023), Nucleotide Transformer (NT) (Dalla-Torre et al., 2025), GENA-LM (Fishman et al., 2025), and GEN-ERator (Wu et al., 2025), which employ BERT-like or Llama-like architectures with tokenization schemes like Byte-Pair Encoding (BPE) or non-overlapping k-mer (novlp-kmer as abbreviation)- where each token corresponds to $k$ consecutive non-overlapping nucleotides. Aims to the specific demands of long-context modeling and single-nucleotide resolution, models such as HyenaDNA (Nguyen et al., 2023), Caduceus (Schiff et al., 2024), and Evo (Brixi et al., 2025) incorporate linear attention mechanisms for efficient long sequences processing while utilizing single-nucleotide tokenization (i.e., {A, T, C, G} as discrete tokens) to preserve base-level sensitivity. Although overlapping k-mer (ovlp-kmer as abbreviation) tokenization also offers single-nucleotide resolution with theoretically better local aggregation and richer vocabulary ($4^k$ size) through its convolution-like overlapped segmentation, it introduces inherent information leakage and requires specialized masking strategies during MLM training (Figure 2), leading to its limited adoption since initial exploration in DNABERT (Ji et al., 2021). Given that parameter-efficient models like HyenaDNA still trail standard BERT-like gLMs across various application (Figure 1), finer-grained tokenization and architectural designs present a promising perspective for bridging this performance gap.

Towards the frontier of parameter-efficient long-context gLM for general DNA modeling, we combine the overlapping k-mer with state-of-the-art Gated DeltaNet (Yang et al., 2024a) to formulate a hybrid gLM model with unified CLM task definition, named HGDNA. The causal masking naturally eliminates bidirectional information leakage in overlapping tokenization while ensuring all tokens are predicted under single nucleotide space, where hybrid backbone ensure its generalizability and efficiency across various context length, with gene-annotated corpus and species classification auxiliary task enable fluence transfer between CLM pretraining and sequence-level prediction. Comprehensive evaluations across 50+ classification benchmarks, zero-shot embedding tasks, and generation tasks demonstrate HGDNA's robust capabilities both in short and long range processing, while revealing the potential of overlapping k-mer and hybrid linear attention for genomic modeling.

## 2 BACKGROUND

**Genomic Language Models** With the noticeable advancement in self-supervised pretraining under massive unlabeled corpus (Devlin et al., 2019; Brown et al., 2020), the domain of genomic analysis, which contains rich sequencing data, is naturally compatible and ready for the NLP-based models to mining the inner short-range and long-range interactions between nucleotides and contigs. In the very beginning, DNABERT (Ji et al., 2021) utilizes the vanilla BERT-based architecture

with overlapping k-mer tokenization ($k \in \{3, 4, 5, 6\}$) to handle the various downstream regulators identification tasks. Nucleotide Transformer family (Dalla-Torre et al., 2025) provides a comprehensive analysis on the scaling-up pretraining in DNA language, with totally 174B-bps multi-species genomic corpus and non-overlapping 6-mer tokenization. DNABERT-2 (Zhou et al., 2023) first introduce the statistically-based BPE tokenization (Sennrich et al., 2016) into gLM, demonstrating a well balance between computational efficiency and overall performance. Evo (Nguyen et al., 2024), Evo2 (Brixi et al., 2025), and GENERator (Wu et al., 2025) provide the paradigm for large-scale autoregressive genomic language pretraining. When turns to the long sequences friendly architectures, HyenaDNA (Nguyen et al., 2023) and Caduceus (Schiff et al., 2024) proofs the effectiveness of full attention-free models in genomic data learning. Both of them achieve meaningful explorations and unique insights in the realm of genomic foundation models.

**Effective Linear Attention with Data-dependency Memory Management**   The crucial component of recent powerful LLMs lies in their high-throughput attention mechanisms (Vaswani et al., 2017; Dao, 2023), which applies independent token-wise interactions through matrix multiplication and softmax. Considering an input sequence $S \in \mathbb{R}^{L \times D}$ with $L$ tokens and $D$ hidden dimension, the standard self-attention returns the output $O \in \mathbb{R}^{L \times D}$ with the same size through:

$$Q, K, V = S\mathbf{W}_Q, S\mathbf{W}_K, S\mathbf{W}_V \tag{1}$$

$$O = \text{softmax}\left((QK^\top) \odot M\right) V \tag{2}$$

where $M \in \{-\inf, 1\}^{L \times L}$ denotes the causal mask for autoregressive modeling. During inference, new token $s_t$ in $t$-th step is required to compute with the previous tokens $S_{[0:t]}$ to generate the $(t+1)$-th step output in the recurrent form:

$$q_t, K_{[0:t]}, V_{[0:t]} = s_t\mathbf{W}_Q, S_{[0:t]}\mathbf{W}_K, S_{[0:t]}\mathbf{W}_V \tag{3}$$

$$o_t = \text{softmax}\left(q_t K_{[0:t]}^\top\right) V_{[0:t]} = \frac{\sum_{i=0}^t \exp(q_t k_i^\top) v_i}{\sum_{i=0}^t \exp(q_t k_i^\top)} \tag{4}$$

Starting with the recurrent form self-attention, linear attentions replace the token-wise softmax reduction to a more simplified kernel function (Katharopoulos et al., 2020) or even the identity mapping without rescaling (Qin et al., 2022), which can be formulated as:

$$o_t = \sum_{i=0}^t (q_t k_i^\top) v_i = q_t \underbrace{\sum_{i=0}^t (k_i^\top v_i)}_{H_t \in \mathbb{R}^{D \times D}} \tag{5}$$

where the $H_t = H_{t-1} + (k_t^\top v_t)$ is the key component of linear attention kernels, representing the matrix-valued memory in recurrent modeling.

Although linear attentions reformulate standard attention in such simplify form, the recurrent form still suffers from inefficient training and long-range modeling, which further motivate the chunk-wise parallel processing (Hua et al., 2022) and finer-grained memory update rule (Dao & Gu, 2024; Yang et al., 2024b;a; Peng et al., 2024). The chunk-wise parallel splits sequence into chunks with length $C$ and computes each chunk's outputs by inter-chunk memory update (recurrent) and intra-chunk ouput computation (parallel), and processing the whole sequence in $\mathcal{O}(LCD + LD^2)$ time complexity:

$$H_{[t]} = H_{[t-1]} + K_{[t]}^\top V_{[t]} \tag{6}$$

$$O_{[t]} = \underbrace{Q_{[t]} H_{[t]}}_{\text{inter-chunk}} + \underbrace{\left(Q_{[t]} K_{[t]}^\top \odot M\right) V_{[t]}}_{\text{intra-chunk}} \tag{7}$$

where $H_{[t]} = H_{[tC:(t+1)C]}$ denotes the matrix value in the $t$-th chunk. For the memory update rules, Mamba2 (Dao & Gu, 2024) introduces data-dependency memory decay as $H_t = \alpha_t H_{t-1} + k_t^\top v_t$, the recent Gated DeltaNet (Yang et al., 2024b;a) rather combines delta update rule with Mamba2's decay mechanisms, which can be formulated as follows:

$$H_t = H_{t-1} - k_t^\top \underbrace{k_t H_{t-1}}_{\text{old value}} + k_t^\top \underbrace{\left(\beta_t v_t + (1 - \beta_t)\left(k_t H_{t-1}\right)\right)}_{\text{new value}} \tag{8}$$

$$\rightarrow \alpha_t \left(I - \beta_t k_t^\top k_t\right) H_{t-1} + \beta_t k_t^\top v_t \tag{9}$$

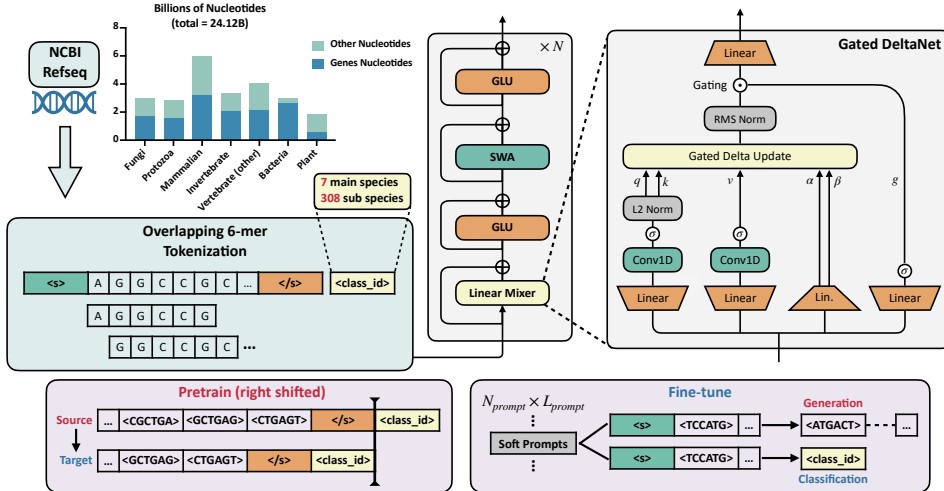

Figure 3: The overview of proposed HGDNA, with sliding window attention (SWA) and Gated DeltaNet as backbone. HGDNA curate 13.69B-bps gene-annotated corpus from 24.12B-bps multi-species reference genomes in NCBI, which includes 7 taxonomic groups. Each sequence is tokenized by overlapping 6-mer and followed by a corresponded species class token as auxiliary training objective. During pretraining and fine-tuning, HGDNA follows a unified next token prediction manner, which samples shared class tokens as target during fine-tuning

where $I$ denotes an identity matrix, $\alpha_t \in (0, 1)$ is the memory decay factor in Mamba2, the "old value" and "new value" denote the erasing and writing mechanisms in DeltaNet's online key-value retrieval with "writing strength" $\beta_t \in (0, 1)$ (Yang et al., 2024a). The better-organized memory management significant reduces the gap between standard attention and linear attention mechanisms in common-sense sequence modeling and long-range retrieving.

## 3 METHOD

As the overall pipeline illustrated in Figure 3, HGDNA is a hybrid causal gLM with combination of Gated DeltaNet (Yang et al., 2024a) and sliding window attention (SWA) (Dao, 2023; Su et al., 2024) aiming to balance short-range capabilities and long-context efficiency, thus can finish pretraining on single NVIDIA A100-40G GPU. The application of overlapping k-mer further boost its potential in functional elements identification. Besides the aforementioned architecture, HGDNA adopts multi-species gene corpus with auxiliary species classification task and shared classification tokens for combining the causal token prediction and sequence-level aggregation. The following sections will present the detailed description of the advantage of HGNDA's designs.

### 3.1 WHEN OVERLAPPING TOKENIZATION MEETS CLM

As noted in Section 1, overlapping tokenization benefits from a scalable vocabulary and its convolution-like strategy, thus providing superior potential in scalability and sequence separability compared to single-nucleotide tokenization (a simplified proof is available in Appendix C). However, the overlapping nature of tokens inevitably introduces information leakage during MLM pretraining, thereby compromising pretraining effectiveness and stability. In this section, we focus on demonstrating the inherent advantages of overlapping tokenization in DNA sequence classification and illustrate how CLM effectively mitigates the issues arising from overlapping scheme.

**CLM Provides Stable Loss Convergence** As illustrated in Figure 2, overlapping tokenization naturally divides the pretraining process into two distinct stages. Initially, the model predicts over a vocabulary of size $4^k$, then progressively learns the overlapping pattern, ultimately reducing the prediction space back to the single nucleotide options. This behavior is also clearly demonstrated in Figure 4: when using 6-mer, the loss drops rapidly from approximately $\log(1/4^6)$ to around

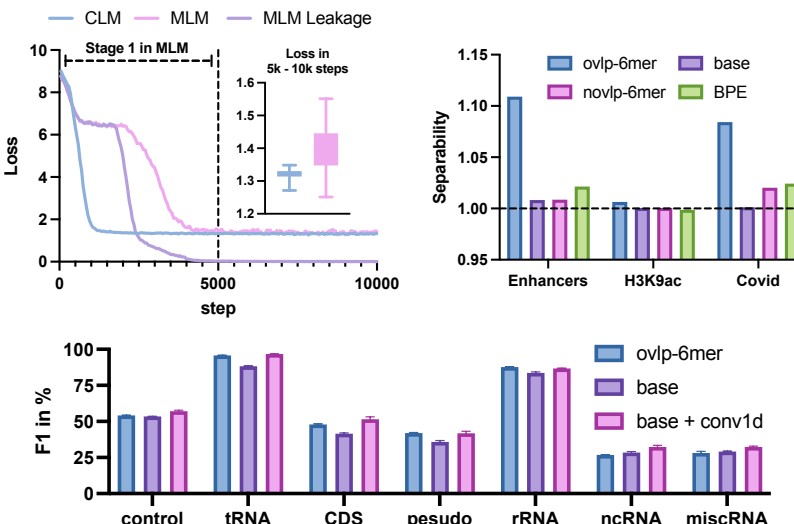

Figure 4: **(Upper Left)** Pretraining loss curves in different pretraining objectives, where "MLM Leakage" denotes without continuous masking. **(Upper Right)** Sequence separability for different tokenization, a value greater than 1.0 indicates that difference between inter-class is larger than intra-class. **(Lower)** XGBoost gene type classification results for different tokenization, embeddings for XGBoost are directly extracted from one-hot tokenized sequence and mean-pooling

$\log(1/4)$. Specifically, the MLM loss exhibits a step-wise decline in the first stage, with convergence notably slower than CLM. After transforming to the 2nd stage, continuous masking in MLM leads to unstable loss convergence, whereas non-contiguous masking results in severe information leakage, driving the loss quickly to zero. These results indicate that CLM-based pretraining more effectively addresses the convergence challenges posed by overlapping tokenization, thereby better leveraging its inherent advantages.

**Overlapping Tokenization Boosts Sequence Separability** Compared to base tokenization, overlapping k-mer inherently incorporates local aggregation, which is theorized to enhance sequence classification capability. To evaluate such advantage, several sequence classification datasets are sampled from the GUE Benchmark (H3K9ac, Covid) (Zhou et al., 2023) and the Nucleotide Transformer Benchmark (Enhancers) (Dalla-Torre et al., 2025). We computed the average intra-class similarity $\mathrm{Sim}_{\mathrm{intra}}$ and inter-class similarity $\mathrm{Sim}_{\mathrm{inter}}$ between tokenized sequences within each dataset using Levenshtein distance, and defined Separability metric as $\mathrm{Sim}_{\mathrm{intra}} / \mathrm{Sim}_{\mathrm{inter}}$. Results in Figure 4 confirm that overlapping k-mer inherently improve sequence separability. Furthermore, we introduce gene classification tasks from Gener Tasks (Wu et al., 2025) by building an XGBoost classifier fed with one-hot encoded token sequences processed through a randomly initialized linear layer and mean pooling. As also shown in Figure 4, overlapping tokenization outperforms in identifying control regions, tRNA, CDS, pseudo genes, and rRNA elements, achieving F1 scores comparable to the combination of base tokenizer with additional Conv1D layer. Together, these results demonstrate that overlapping k-mer provide substantial benefits at the tokenization stage even prior to the introduction of learnable parameters.

## 3.2 Bridging Pretraining and Fine-tuning via Species Classification

Beyond carefully selected tokenization and pretraining objective, in order to maximize HGDNA's capability under limited parameters, the multi-species gene corpus with corresponded species labels are introduced as the main target and auxiliary tasks for pretraining, which further effectively bridges the gap between CLM and sequence-level prediction.

**Pretraining** In the pretraining stage, following the insights from GENERator (Wu et al., 2025) and Evo2 (Brixi et al., 2025), HGDNA extracts 13.6B-bps gene-annotated fragments from 24.1B-bps reference genomics as the training data for reducing repetitive and redundant corpus. Further-

more, to incorporate natural language-like explicit context learning in the pretraining stage, the taxonomic labels are concatenated to the end of each input sequence to align with broader downstream sequence-level classification. The aforementioned strategies yield the following formulation of pretraining inputs:

$$< \text{start token} >< \text{gene}_1 > \mid < \text{gene}_2 > \mid \cdots \mid < \text{gene}_n >< \text{end token} >< \text{class token} > \quad (10)$$

where $< \text{gene} >$ denotes the gene fragments from the same species, $\mid$ denotes the delimiter between each gene, $< \text{class token} >$ denotes several pre-defined tokens in vocabulary $\mathbb{V} = \{\mathbb{V}^n, \mathbb{V}^c\}$ for sequence-level classification, with $\mathbb{V}^n$ for normal tokens (e.g., 6-mer tokens and special tokens) and $\mathbb{V}^c$ for class tokens, the overall size of $\mathbb{V}^c$ is set to 1,024, indicating that HGDNA can support 1,024-classes classification during pretraining and fine-tuning. The whole sequence is fed into the vanilla cross-entropy for token-wise loss.

**Fine-tuning** After initialized by species classification auxiliary task during pretraining, the whole $\mathbb{V}^c$ are shared as classification labels for downstream adaption. For sequence classification, unlike naive [CLS] token extraction and mean-pooling, HGDNA follows Chen et al. (2023) to perform a single-step generation for sampling the next classification tokens according to provided contexts. As for sequence design, HGDNA further introduce the learnable soft prompts (Lester et al., 2021; Nguyen et al., 2023) for instructed generation on pure gLM. Considering an input sequence $S \in \mathbb{R}^{L \times D}$ with its label $i$ in downstream tasks, HGDNA selects $P_i \in \mathbb{R}^{L' \times D}$ from a set of prompts $\{P_1, \cdots, P_N\} \in \mathbb{R}^{N \times L' \times D}$ and add to the beginning of $S$ to guide the generation process, where $N$ and $L'$ denotes the types and length of prompts, respectively. We further employ the first prompt $P_1$ for each sequence during classification and regression for finer-grained tuning.

## 4 EXPERIMENTS

### 4.1 BASELINES

As a long context-friendly gLM with 30M parameters, we introduce two types of gLMs as baselines: (1) parameter-efficient gLMs with scale and training budget similar to HGDNA, e.g., HyenaDNA (Nguyen et al., 2023) and Caduceus-Ps (Schiff et al., 2024). (2) standard BERT-like gLMs with full-attention and larger training budget, e.g., DNABERT-2 (Zhou et al., 2023), NTv2 family (Dalla-Torre et al., 2025), and GENA-LM (Fishman et al., 2025). The fundamental purpose of HGDNA is to achieve similar or even superior performance to the state-of-the-art BERT-like gLMs that requires 10-50x more training tokens while maintaining high efficiency in long-range sequence processing (Table 1).

Table 1: Total parameters, corpus size (in bps), and pretraining tokens for each model, with † denotes data augmentation from 1000-Genomes Project

| Methods | Params. | Corpus | Budgets |
|---|---|---|---|
| NTv2-100M | 98M | 135B | 300B |
| NTv2-500M | 498M | 135B | 1,000B |
| DNABERT-2 | 117M | 32.5B | 262B |
| GENA-LM | 110M | 3.2B† | 275B |
| Caduceus-Ps | 8M | 3.2B | 52B |
| HyenaDNA | 6M | 3.2B | 5B |
| HGDNA | 30M | 13.6B | 19B |

### 4.2 SHORT-RANGE BENCHMARKS

**GUE Benchmark** GUE (Zhou et al., 2023) is a standard short-range sequence classification benchmark containing 28 datasets across 7 main tasks and 4 species, with sequence length from 70-bps to 1,000-bps. Except Covid variant classification (CVC) is evaluated by F1-score, the remaining splice site prediction (SSP), promoter and core-promoter detection (PD & CPD), transcription factors prediction in human and mouse (TFH & TFM), and epigenetic marker prediction (EMP) are evaluated by Matthews Correlation Coefficient (MCC). As the evaluation results in Table 2, HGDNA provides a significant improvement over the state-of-the-art gLMs, especially for the EMP tasks, highlighting its generalizability across multi-species short-range tasks.

**NT (revised) Benchmark** The revised version of NT Benchmark is a human-specific benchmark including 18 datasets across epigenetic markers prediction, enhancers detection (ED), promoters detection, and splice site prediction, with sequence lengths from 300-bps to 1,000-bps and metric set to MCC. We further split chromosome 10 (chr10) data in each training set as held-out validation set.

Table 2: Averaged MCC/F1 performance (in %) on GUE and Nucleotide Transformer (revised) benchmarks across 3 runs, performance ranked in top-1 and top-2 are highlighted in **bold** and underline, respectively. "–" denotes models fail to convergence on the dataset across 3 runs

| Models | GUE Benchmark | | | | | | | | NT Benchmark | | | | |
|---|---|---|---|---|---|---|---|---|---|---|---|---|---|
| | EMP | TFM | CVC | TFH | PD | CPD | SSP | Avg. | EMP | ED | PD | SSP | Avg. |
| NTv2-100M | 54.03 | 65.63 | 61.93 | 64.16 | 85.26 | 71.59 | 86.00 | 64.56 | 54.58 | 55.86 | 82.15 | 96.35 | 66.28 |
| NTv2-500M | 57.63 | 71.59 | 63.47 | 67.36 | **91.39** | **74.74** | **90.49** | **68.69** | 56.74 | **56.53** | **84.01** | **97.06** | **67.98** |
| DNABERT-2 | 56.02 | 71.00 | 71.02 | **68.95** | 79.56 | 69.61 | 86.15 | 66.59 | **56.82** | 52.41 | 77.85 | 83.31 | 64.25 |
| GENA-LM | 47.01 | 64.69 | – | 64.38 | 81.12 | 67.16 | 76.55 | 60.62 | 56.01 | 51.78 | 76.58 | 80.86 | 63.11 |
| HyenaDNA | 48.32 | 65.47 | 46.48 | 60.09 | 77.31 | 69.60 | 83.20 | 60.05 | 52.63 | 50.62 | 78.95 | 82.24 | 61.73 |
| Caduceus-Ps | 53.99 | 69.74 | 68.64 | 67.22 | 75.65 | 67.05 | 83.51 | 64.46 | 51.67 | 47.57 | 75.22 | 78.63 | 59.63 |
| HGDNA | **68.40** | **72.33** | **72.50** | 64.78 | 86.50 | 72.66 | 90.02 | **71.77** | 55.63 | 53.64 | 80.44 | 95.11 | 66.12 |

Table 3: Averaged MCC performance (in %) and standard deviation on several long-range benchmarks across 3 runs. "*" denotes to apply chunk-by-chunk processing for long sequences

| Models | Gener | | Vertebrate | | | Zero-shot Regulators Classification | | |
|---|---|---|---|---|---|---|---|---|
| | gene | taxonomic | 1k | 16k | 32k | PDH | PDM | EMP |
| NTv2-100M | 62.59 (0.45) | 98.02 (0.11)* | 52.97 (0.30) | 95.65 (0.17) | 99.62 (0.07)* | 83.61 (0.00) | 55.45 (0.00) | 10.30 (0.00) |
| NTv2-500M | **64.18 (0.22)** | – | 55.91 (0.39) | **97.23 (0.21)** | 81.37 (1.31)* | 84.67 (0.00) | 56.77 (0.00) | 11.62 (0.00) |
| DNABERT-2 | 60.42 (0.39) | 97.44 (0.05)* | **56.48 (0.32)** | 96.59 (0.19) | 99.74 (0.26)* | 83.95 (0.00) | 51.44 (0.00) | 8.73 (0.00) |
| HyenaDNA | 56.51 (0.39) | 95.29 (0.17) | 36.65 (0.34) | 65.80 (0.12) | 80.44 (0.08) | 84.05 (0.00) | 57.78 (0.00) | 3.58 (0.00) |
| Caduceus-Ps | 56.39 (0.43) | 96.61 (0.16) | 46.47 (0.35) | 96.98 (0.18) | 99.11 (0.14) | **86.28 (0.00)** | 60.77 (0.00) | 3.58 (0.00) |
| HGDNA | 63.17 (0.51) | **98.30 (0.30)** | 53.23 (0.41) | 97.09 (0.17) | **99.74 (0.15)** | 85.47 (0.00) | **64.26 (0.00)** | **13.88 (0.00)** |

As the results provided in Table 2, HGDNA achieves competitive performance to the NTv2 family while utilizing only 5% of its pretraining budget, especially for the SSP task that only HGDNA and NTv2 achieve near-perfect performance, significantly surpassing other baselines.

### 4.3 Long-range Benchmarks

**Gener Tasks**  To explore HGDNA's potential in long sequence understanding, the Gener Task in GENERator (Wu et al., 2025) is introduced to access models' performance on near 100k-bps context. The benchmark contains two distinct tasks: (1) gene classification, which evaluates models on annotating gene types from multi-species genomic corpus with sequence length from 67-bps to 5k-bps; (2) taxonomic classification, which requires model to process 96k-bps long multi-species sequences and predict their taxonomic groups. As the results in Table 3, HGDNA achieves the best performance in multi-species sequences classification, whereas the models like DNABERT-2 require chunk-by-chunk processing to pass the fine-tuning stage, and NTv2-500M even unable to converge under limited chunk size due to memory limitation.

**Vertebrate Species Classification**  The vertebrate species classification task is another long-context benchmark adopted from Nguyen et al. (2023), which chunks sequences from 5 vertebrate species to form 5-classes classifications in 1k, 16k, and 32k context lengths, requiring models to discriminant the species-wise divergence and variant. As results in Table 3, HGDNA achieves top-1 and top-2 performance when context exceeds 16k, the visualization of embedding (Figure 6) also reveals HGDNA's superior discrimination capability over remaining parameter-efficient models like Caduceus-Ps. When move to long-context efficiency (Figure 5), HGDNA still maintains its superiority over baselines, with over 25x speedup compared to NTv2-500M when

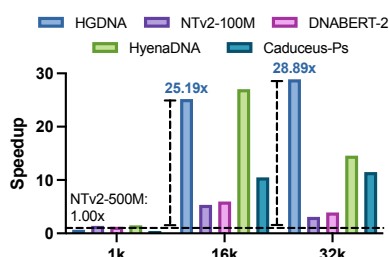

Figure 5: Inference throughput on vertebrate species classification task, with baseline set to NTv2-500M

context exceeds 16k, which also significantly surpass HyenaDNA and Caduceus-Ps when context exceeds 32k.

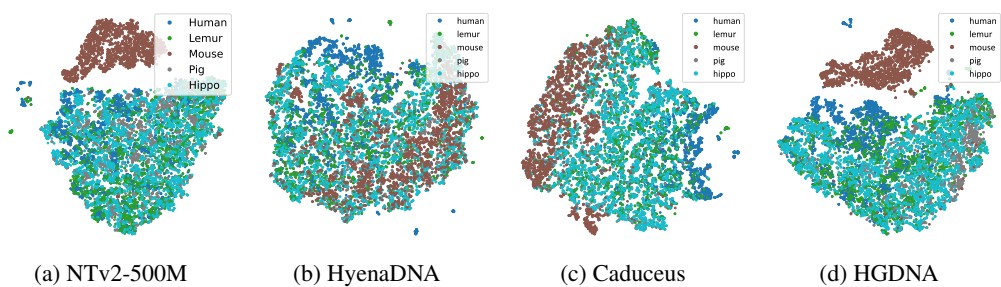

| (a) NTv2-500M | (b) HyenaDNA | (c) Caduceus | (d) HGDNA |

Figure 6: t-SNE visualization for zero-shot vertebrate species embeddings

**Zero-shot Regulators Classification** Compared to task-by-task fine-tuning, zero-shot embedding presents a more efficient solution for downstream tasks with gLMs. Herein, we introduce human promoter detection (PDH), multi-species promoter detection (PDM), and multi-species EMP zero-shot embedding tasks (Feng et al., 2024), which contains 4+4+6 datasets, with context length exceeds 2k-bps. For all pretrained models, mean-pooled sequence embeddings are directly used to train an XGBoost classifier for evaluation on the test set. As shown in Table 3, while HGDNA's embeddings for human promoters are slightly inferior to Caduceus-Ps that trained purely on human data, HGDNA significantly outperforms all baselines on the multi-species data. This result underscores HGDNA's capability in multi-species tasks.

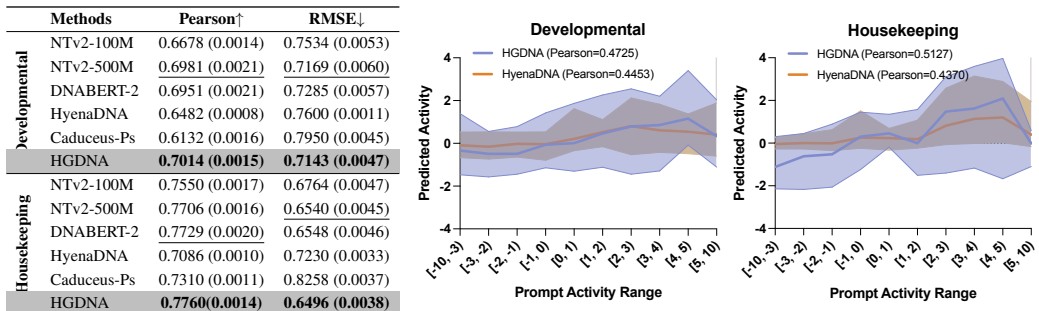

| | Methods | Pearson↑ | RMSE↓ |
|---|---|---|---|
| Developmental | NTv2-100M | 0.6678 (0.0014) | 0.7534 (0.0053) |
| | NTv2-500M | 0.6981 (0.0021) | 0.7169 (0.0060) |
| | DNABERT-2 | 0.6951 (0.0021) | 0.7285 (0.0057) |
| | HyenaDNA | 0.6482 (0.0008) | 0.7600 (0.0011) |
| | Caduceus-Ps | 0.6132 (0.0016) | 0.7950 (0.0045) |
| | **HGDNA** | **0.7014 (0.0015)** | **0.7143 (0.0047)** |
| Housekeeping | NTv2-100M | 0.7550 (0.0017) | 0.6764 (0.0047) |
| | NTv2-500M | 0.7706 (0.0016) | 0.6540 (0.0045) |
| | DNABERT-2 | 0.7729 (0.0020) | 0.6548 (0.0046) |
| | HyenaDNA | 0.7086 (0.0010) | 0.7230 (0.0033) |
| | Caduceus-Ps | 0.7310 (0.0011) | 0.8258 (0.0037) |
| | **HGDNA** | **0.7760(0.0014)** | **0.6496 (0.0038)** |

Figure 7: (**Left**) Regression results in DeepSTARR's drosophila melanogaster enhancer activity datasets. (**Right**) Activity distribution of generated enhancers. Each sequence is generated based on the soft prompt in x-axis, with y-axis denotes to its activity predicted by the regression model

## 4.4 INSTRUCTION-BASED DESIGN OF ENHANCERS WITH SPECIFIC ACTIVITY

To further evaluate HGDNA's potential in sequence generation, we adopt DeepSTARR's drosophila melanogaster enhancer activity datasets (de Almeida et al., 2022) to fine-tune models for generating new enhancers according to the provided activity profiles, and trains a distinct regression model to evaluate the activity of generated enhancers. Unlike GENERator that employs $< low >$ and $< high >$ tokens for generation guidance (Wu et al., 2025), we discretize developmental and housekeeping activities into 10 distinct labels mapped to corresponding soft prompts, thereby enabling finer-grained control for generating enhancer under specific activity range. Details of the datasets and task settings are available in Appendix B. According to the regression results in Figure 7, HGDNA provides the best Pearson correlation and the minimum RMSE between ground truth and predicted activities, thereby adopted as the activity predictor in the generation task. As for the generation quality, HGDNA generates wider activity range sequences compared to HyenaDNA, with higher Pearson correlation to provided prompts, which confirm the potential of fine-grained instructed sequence design in both HGDNA and broader autoregressive models.

## 4.5 ABLATION STUDY

**Macro Designs** As illustrated in Figure 8, the key design of HGDNA can be divided into 4 chapters. From the start point with vanilla DNABERT-2 (BPE + MLM) architecture, overlap-

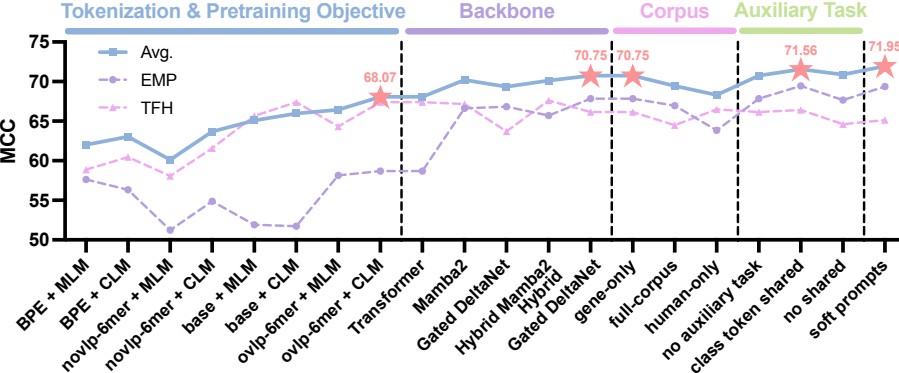

Figure 8: Ablation studies over tokenization, pretraining objectives, backbone selection, training corpus, auxiliary task transferring, and other micro designs on GUE benchmark. All models are pretrained with 60% budget of the vanilla HGDNA with the same 30M parameters scale

ping tokenization with CLM pretraining provides the top-ranked boost (Avg. +6.04) in both multi-species EMP task and human-oriented TFH task, which also surpass the vanilla single nucleotide tokenization by 2.97 averaged MCC. As for the backbone and training corpus, the hybrid Gated DeltaNet is notably benefiting EMP task, with averaged 9.12 MCC boosting compared to full-attention, while extracting the gene-only region as corpus further provides averaged 1.29 MCC boosting. Another key design of HGDNA is lying into the species classification auxiliary task and shared class tokens, which jointly enhance overall MCC by 0.81, replacing the CLM-like shared token generation to naive new-initialized [CLS] projection will lead to suboptimal effect.

**Micro Designs Tuning** Beyond macro designs, the k-mer size and memory management strategies are also forming the advantages from tokenization and linear attention backbone. As results posted in Table 4, different $k$ can significantly affect HGDNA's capabilities over various tasks, with $k < 5$ benefit to short-range TFM and CPD tasks, and larger $k$ can boost the performance on longer sequence, e.g., EMP and CVC. As for the brand-new designs in recent linear attention algorithms, both explicit memory decay and KV retrieval optimization can improve the capabilities on 96k-bps long sequences, with Delta update rule plays a more important role in the DNA modeling context.

Table 4: (**Upper**) Ablation studies on $k$ size, with backbone set to HGDNA without auxiliary task. (**Lower**) Ablation studies on linear attention designs for 96k-bps taxonomic classification, with all baselines set to 3-layers pure linear attention and trained from scratch

| $k$ | EMP | TFM | CVC | TFH | PD | CPD | SSP | **Avg.** |
|---|---|---|---|---|---|---|---|---|
| 3 | 58.07 | **75.23** | 69.62 | 66.18 | 80.67 | **73.13** | 89.98 | 68.17 |
| 4 | 58.92 | 72.16 | 71.89 | **67.22** | 82.33 | 72.48 | 88.01 | 68.23 |
| 5 | 65.88 | 74.27 | 71.80 | 66.97 | 77.02 | 70.89 | **90.74** | 70.40 |
| 6 | 67.83 | 70.20 | 72.71 | 66.14 | **83.20** | 69.89 | 88.95 | **70.75** |
| 7 | **69.98** | 70.95 | **73.28** | 66.56 | 76.47 | 67.12 | 85.34 | 70.66 |

| | **Memory Decay** | **KV Retrieval** | **MCC** |
|---|---|---|---|
| RetNet[1] | data-independent | – | 76.65 (0.47) |
| Mamba2[1] | data-dependent | – | 77.35 (0.41) |
| DeltaNet[1] | – | ✓ | 78.56(0.52) |
| Gated DeltaNet[1] | data-dependent | ✓ | **78.76 (0.45)** |

## 5 CONCLUSION

In this study, we highlight the potential of overlapping k-mer tokenization in autoregressive pre-training and present HGDNA–a hybrid Gated DeltaNet model pretrained on species-labeled, gene-specific multi-species corpus. Leveraging linear-time complexity and the unified autoregressive pipeline, HGDNA outperforms existing linear-complexity gLMs and in both long-context throughput and various downstream applications, and further align to the state-of-the-art gLMs with only 5% training budget, while retaining potential for further instruction-guided sequence generation.

---

[1]Implement by flash-linear-attention package (Sun et al., 2023; Dao & Gu, 2024; Yang et al., 2024b;a)

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

## A    DETAILS OF PRETRAINING IMPLEMENTATION

**Pretraining Corpus**    We developed an automated pipeline to randomly retrieve annotated reference genomes from the NCBI RefSeq FTP server[1]. Our data collection strategy involved predefined categories and sizes: "fungi" (3.0GB), "protozoa" (3.0GB), "vertebrate_mammalian" (6.0GB), "invertebrate" (3.0GB), "vertebrate_other" (3.0GB), "bacteria" (3.0GB), and "plant" (3.0GB). For each taxonomic category (except "vertebrate_mammalian", which is specifically targeted to human and mouse data), we first downloaded the corresponding "assembly_summary.txt" file, which contains metadata and download links for all available genomes in current taxonomic group. We then randomly selected genomes from the summary file until reaching the predefined size thresholds. The final datasets, with both raw genome sequences (.fna) and annotation files (.gff), are acquired through parallelized wget downloads.

After downloading the raw genomic corpus, we process the raw genomic data by first filtering sequences based on their annotation information, then constructing the final pretraining corpus in FASTA format. Specifically, we: (1) extract genomic fragments which GFF types are annotated as "gene"; (2) concatenated all gene fragments from the same species using "—" as delimiter; and (3) add a header containing species metadata to each sequence. Ahead on the coarse-grained 7 taxonomic groups and extremely imbalanced scales across 887 species types, we implemented a grouping strategy, which treats species sharing the same prefix (e.g., "Penicillium_alfredii" and "Penicillium_oxalicum") as the same group. This approach finally yields 308 species labels for the pretraining phase of HGDNA.

Table 5: Detailed statistics of the multi-species pretraining corpus used by HGDNA

| NCBI Refseq Category | Num. of Species | Genes Nucleotides / Total Nucleotides (bp) |
|---|---|---|
| Fungi | 91 | 1.66B / 2.97B |
| Protozoa | 81 | 1.51B / 2.80B |
| Mammalian | 2 | 3.20B / 6.01B |
| Invertebrate | 6 | 2.05B / 3.36B |
| Vertebrate (other) | 3 | 2.13B / 4.09B |
| Bacteria | 699 | 2.59B / 2.99B |
| Plant | 5 | 0.54B / 1.88B |
| All | 887 | 13.69B / 24.12B |

**Tokenization and Vocabulary**    Similar to the non-overlapping 6-mer tokenization in Nucleotide Transformer (NT), the overlapping 6-mer also contains a fixed 6-mer vocabulary in sized $4^6$, which corresponding to all possible combinations of 6 nucleotides under randomly arrangement. In addition to the fixed 4096 6-mer tokens, single nucleotide tokens, and special tokens which form the entire normal vocabulary $\mathbb{V}^n$, we further define 1024 classification tokens ($<$ class0 $>$ through $<$ class1023 $>$) in the class vocabulary $\mathbb{V}^c$, which is adopted as species labels during pretraining, and class labels during downstream sequence classification tasks. The above tokens ultimately form HGDNA's vocabulary $\mathbb{V}$ with the size of 5130.

For other tokenization used in ablation studies, we employ the same special tokens set as the aforementioned overlapping 6-mer tokenization. The vocabulary of BPE is directly downloaded from DNABERT-2, with minor modifications to make compatible with the same special tokens set (e.g., delimiter between genes).

**Sequence Length Warm-up during Pretraining**    Following Li et al. (2022), we adopt a linear sequence length warm-up (SLW) strategy during the initial step of pretraining for stability and faster convergence, which can be formulated as follows:

$$L_t = L_{min} + (L_{max} - L_{min}) \times \min\left(\frac{t}{T_{SLW}}, 1\right) \tag{11}$$

---

[1]https://ftp.ncbi.nlm.nih.gov/genomes/refseq/

where $L_t$ denotes the target sequence length at step $t$, $L_{max}$ denotes the upper bound of sequence length, $L_{min}$ denotes the lower bound of sequence length, $T_{SLW}$ denotes the total steps for sequence length warm-up schedule. Furthermore, due to the hard thresholds of maximum tokens in each batch ($N_{token}$), the batch size $B$ is descent throughout the warm-up phase:

$$B_t = \min\left(B_{max}, \lfloor \frac{N_{token}}{L_t} \rfloor\right) \tag{12}$$

where $B_{max}$ denotes the maximum batch size. The above warm-up settings enables HGDNA to achieve efficient pretraining convergence within predictable computational budget.

**Pretraining Settings** For the pretraining of HGDNA and other models for ablation studies, we adopt the hyperparameters settings in Table 6. The final HGDNA model includes 3 hybrid Gated DeltaNet layers, with 1 Gated DeltaNet, 1 sliding window attention, and 2 GLU in each layer. All the pretraining stages are finished in a single NVIDIA A100-40GB GPU. It is worth noting that the base of rotary embedding is expanded to 1M (Liu et al., 2023), which enables HGDNA to fit the ultra-long sequence in downstream tasks, even though the maximum sequence length in pretraining stage is limited to 2,048-bps. Additionally, to maintain the diversity of pretraining data and model's capability for handling variable length sequences, 50% of the sequences are reversed and mapped to its complement base-pairs (e.g., "A" to "T", "C" to "G", etc.), where 10% sequences in each batch are adopted with the sequence length ranges in $[L_{min}, L_{max}]$.

To reaching the optimal hardware utilization, the Gated DeltaNet is implemented from flash linear attention (fla) library[2], which provides a high-performance triton version of chunk-wise processing (Tillet et al., 2019). Mamba2[3] and flash attention 2[4] are build from their source code.

**Details of Forward Process in Gated DeltaNet** Following Section 2, the key design of Gated DeltaNet is referred to its memory management mechanisms. Consider an input token $s_t \in \mathbb{R}^D$ at current step $t$, the overview of forward pass is similar to Figure 3, with prepare stage for generating query, key, value, $\alpha$, $\beta$, and final gating $g$ as follows:

$$q_t, k_t, v_t = \text{Conv1D}(s_t\mathbf{W}_Q), \text{Conv1D}(s_t\mathbf{W}_K), \text{Conv1D}(s_t\mathbf{W}_V) \tag{13}$$
$$\alpha_t = \exp\left(\text{SoftPlus}(s_t\mathbf{W}_\alpha) \odot \mathbf{b}_\alpha\right) \tag{14}$$
$$\beta_t = \sigma(s_t\mathbf{W}_\beta) \tag{15}$$

After obtaining the stats for current status, the calculation of final output $o_t$ and updated memory $H_t$ can be parameterized as follows according to the implementation in fla library:

$$H'_{t-1} = (H_{t-1} \odot \alpha_t) \tag{16}$$
$$\Delta v_t = \beta_t(v_t - k_t H'_{t-1}) \tag{17}$$
$$H_t = H'_{t-1} + k_t^\top \Delta v_t \tag{18}$$
$$\rightarrow \alpha_t \left(I - \beta_t k_t^\top k_t\right) H_{t-1} + \beta_t k_t^\top v_t \tag{19}$$
$$o_t = \text{SwishGLU}(q_t H_t, \alpha_t \mathbf{W}_g) \tag{20}$$

where $\text{Conv1D}(\cdot)$ denotes a causal convolution operation over sequence dimension, formulated by $\sum_{i=0}^{k-1} w_i * x_{t-i}$ with kernel size $k$ and kernel parameters $w$. $\text{SoftPlus}(\cdot)$ denote the softplus activation function defined as $\log(1+e^x)$, and $\text{SwishGLU}(\cdot)$ denotes the GLU layer with Swish/SiLU activation as $((\sigma(x_1\mathbf{W}_1) \odot x_1\mathbf{W}_1) \odot x_2\mathbf{W}_2)\mathbf{W}_3$. Compare with the formulation in Section 2, $k_t^\top \Delta v_t$ in equation 19 can be further regarded as the gradient of online SGD optimization over key-value regression loss $\nabla_{H_{t-1}} \mathcal{L}_t(H_{t-1})$, where $\nabla_{H_{t-1}} \mathcal{L}_t(H_{t-1}) = \frac{1}{2}\|v_t - k_t H_{t-1}\|^2 = k_t^\top (k_t H_{t-1} - v_t)$, with $\beta_t$ denotes to the learning rate (Yang et al., 2024b).

---

[2]https://github.com/fla-org/flash-linear-attention
[3]https://github.com/state-spaces/mamba
[4]https://github.com/Dao-AILab/flash-attention

Table 6: Hyperparameters settings for pretraining

| Settings | Values |
|---|---|
| Layers | 3 (for hybrid models) or 6 |
| Hidden size | 512 |
| Intermediate size | 2,048 |
| Feedforward type | GLU (Shazeer, 2020) |
| Attention heads | 8 |
| Attention window size | 1,024 (for hybrid models) |
| Gated DeltaNet heads | 6 |
| Eps | 1e-5 |
| Dropout | 0.1 |
| Rotary embedding base | 1e6 |
| Parameters | 30M |
| Optimizer | AdamW (Loshchilov & Hutter, 2018) |
| Optimizer momentum | $\beta_1 = 0.9, \beta_2 = 0.98$ |
| Optimizer weight decay | 0.1 |
| Gradient clipping | 1.0 |
| Total update steps | 200,000 or 120,000 (for ablation studies in Figure 8) |
| Minimum learning rate | 1e-6 |
| Maximum learning rate | 5e-4 |
| Learning rate warm-up steps | 10,000 |
| Learning rate decay | Cosine decay |
| Maximum batch size ($B_{max}$) | 128 |
| Maximum tokens per batch ($N_{token}$) | 98,304 or 65,536 (for ablation studies in Figure 8) |
| Minimum sequence length ($L_{min}$) | 64 |
| Maximum sequence length ($L_{max}$) | 2,048 |
| Sequence length warm-up steps ($T_{SLW}$) | 40,000 |
| Random length rate | 0.1 |
| Training Cost | NVIDIA A100-40G * 14h |

## B    DETAILS OF DOWNSTREAM EVALUATIONS

**Models Setup**    The main target of HGDNA is comparing with HyenaDNA (Nguyen et al., 2023), Caduceus (Schiff et al., 2024), NT (Dalla-Torre et al., 2025), and DNABERT-2 (Zhou et al., 2023) across various downstream tasks. We download and fine-tune the following model checkpoints from Huggingface repositories for comparison:

- NTv2-100M: InstaDeepAI/nucleotide-transformer-v2-100m-multi-species
- NTv2-500M: InstaDeepAI/nucleotide-transformer-v2-500m-multi-species
- DNABERT-2: zhihan1996/DNABERT-2-117M
- GENA-LM: AIRI-Institute/gena-lm-bert-base-t2t
- HyenaDNA: LongSafari/hyenadna-medium-160k-seqlen-hf
- Caduceus: kuleshov-group/caduceus-ps_seqlen-131k_d_model-256_n_layer-16

For other baselines in GUE benchmark, we directly use the fine-tuning results from DNABERT-2 (Zhou et al., 2023). As for the soft-prompt in HGDNA, we concatenate a 64-length tunable tensor for all the inputs during classification and regression fine-tuning, while create a $10 \times 64$ soft-prompt tensor for guiding the model to generate enhancers within 10 activity ranges.

Throughout the fine-tuning process, all models are applied to full fine-tuning with a single NVIDIA A100-40G GPU. Specifically, we employ pytorch-lightning[5] framework with a customized checkpoint ensemble callback to average the top-5 checkpoints and capture the best performance for all models. Each dataset is run 3 times with random seed in $\{17, 18, 19\}$ to obtain the average performance. Specific for HGDNA, which already includes pretrained class tokens into the vocabulary, we fine-tune the model to generate the corresponded class tokens after the end token to form the classification pipeline, with other baselines employing a new initialized projection head to predict the classification results. Since HGDNA only generate a single class token during classification tasks, we employ the bidirectional sliding window attention for capturing neighboring context information from two sides, with window size consistent with generation tasks (e.g., 1,024 tokens before the current step in autoregressive modeling, and 512 tokens from bidirectional neighbors in classification and regression tasks).

**GUE Benchmark**    We directly download GUE benchmark from the source code of DNABERT-2, maintaining the same data split. Following the settings in DNABERT-2, we set batch size to 32, with a 100 steps linear learning rate warm-up to reach the 5e-5 threshold without decay. The update steps are set to 10k for Covid, and 6k for the remaining classification tasks. Detailed evaluation results are available in Table 13.

**NT (revised) Benchmark**    The metadata of NT (revised) is downloaded from Huggingface Datasets[6]. To make compatible with the ensemble callback, we further filter out the chr10 data from the original training set and form a new validation set, the details of modified NT (revised) benchmark is available in Table 7. For the downstream fine-tuning, each dataset is run in the same settings as EMP task in GUE, with 32 batch size and 6k steps. Detailed evaluation results are available in Table 14.

**Gener Benchmark**    The two classification datasets of Gener benchmark are downloaded from Huggingface Datasets[7]. We create a revised version that randomly sample 10% data from the original training set to form a validation set, with detailed statistics in Table 8. Since Gener task includes longer context than GUE and NT (revised) benchmark, we adopt gradient accumulation to maintain the global batch size and updating steps to $32 \times 6000$ for gene classification and $16 \times 3000$ for taxonomic classification, with other settings similar to the NT (revised) benchmark. Notably, neither NTv2-100M nor DNABERT-2 could be fine-tuned on a single A100-40G GPU, even with batch size set to 1 and full bfloat16 training, while the official HyenaDNA implementation will suffer from memory leakage issue in 96k-bps context fine-tuning. To finish the fine-tuning for NTv2-100M, and

---

[5]https://github.com/Lightning-AI/pytorch-lightning

[6]InstaDeepAI/nucleotide_transformer_downstream_tasks_revised

[7]GenerTeam/gener-tasks

918
919
920
921
922
923
924
925
926
927
928
929
930
931
932
933
934
935

Table 7: Detailed statistics of tasks in NT benchmark (revised)(Dalla-Torre et al., 2025). The original benchmark only provides train/test split, with test set in chr20 and chr21, we revised the data split to train/dev/test, with additional dev set in chr10. All tasks are managed in binary classification except 3-classes enhancer types classification

| Tasks | Sequence Length | Data Split (train/dev/test) |
|---|---|---|
| Epigenetic Marks Prediction | | |
|   H2AFZ | | 28,583 / 1,417 / 3,000 |
|   H3K27ac | | 28,520 / 1,480 / 1,616 |
|   H3K27me3 | | 28,505 / 1,495 / 3,000 |
|   H3K36me3 | | 28,690 / 1,310 / 3,000 |
|   H3K4me1 | 1k | 28,554 / 1,446 / 3,000 |
|   H3K4me2 | | 28,430 / 1,570 / 2,138 |
|   H3K4me3 | | 16,553 / 915 / 776 |
|   H3K9ac | | 22,144 / 1,130 / 1,004 |
|   H3K9me3 | | 26,143 / 1,296 / 850 |
|   H4K20me1 | | 28,535 / 1,465 / 2,270 |
| Enhancers Prediction | | |
|   identification | 400 | 28,542 / 1,458 / 3,000 |
|   types classification | | 28,542 / 1,458 / 3,000 |
| Promoters Prediction | | |
|   all | | 28,635 / 1,365 / 1,584 |
|   tata | 300 | 4,844 / 218 / 212 |
|   notata | | 28,632 / 1,368 / 1,372 |
| Splice Site Prediction | | |
|   all | | 28,597 / 1,403 / 3,000 |
|   acceptors | 600 | 28,519 / 1,481 / 3,000 |
|   donors | | 28,582 / 1,418 / 3,000 |

DNABERT-2 in such ultra-long context, we split the input sequences into 16k-bps chunks to generate chunk-level representations, then applies the mean pooling to obtain the final sequence-level states for classification (Figrue 9). For NTv2-500M, the chunk size is required to be 512-bps for training in single A100-40G GPU, which significantly affects its convergence stability and performance.

Table 8: Detailed statistics of tasks in gener benchmark. The original benchmark only provides train/test split, we randomly sample 10% data from the original training set to form the dev set. The gene classification tasks is managed in 7-classes classification, where taxonomic classification is managed in 6-classes classification

| Tasks | Sequence Length | Data Split (train/dev/test) |
|---|---|---|
| Gene Classification | 67 - 4.94k (avg. 381) | 72,886 / 8,098 / 20,244 |
| Taxonomic Classification | 96k | 45,446 / 5,050 / 5,610 |

```
1  def chunk_by_chunk_processing(glm, input_ids, chunk_size):
2      # glm: gLMs from Huggingface
3      # input_ids: tokenized sequences with shape [Batch Size, Sequence
           Length]
4      # chunk_size: length of each chunk
5
6      B, L = input_ids.shape
7      chunks = []
8      for i in range((B + chunk_size - 1) // chunk_size):
9          outputs = glm(input_ids[:, i * chunk_size:(i + 1) * chunk_size])
10         chunks.append(outputs.logits)
11
12     return torch.cat(chunks, dim=1).mean(1)
```

Figure 9: Pytorch-like code snippet of the chunk-by-chunk processing for DNABERT-2, NTv2-100M, and NTv2-500M. The final chunk lists are directly aggregated by mean-pooling since other methods like explicit attention layer provides similar performance

**Vertebrate Species Classification**    The raw sequence data for human, lemur, mouse, pig, hippo are download from NCBI Genomes following the links provided by HyenaDNA[8]. For the standardization of benchmarks and efficient validation, we reconstruct the benchmark by randomly sample fragments from the five species' genomes with pre-defined context length (1k, 16k, and 32k). The training set is sampled from chr1-9, with chr10 as validation set and chr11 as test set for held-out evaluation. Table 9 provides the details about dataset statistics.

For the supervised fine-tuning of species classification, we follow the same hyperparameters settings as NT (revised) benchmark with 32 global batch size across all context settings. Similar as the taxonomic classification in Gener tasks, NTv2-100M and DNABERT-2 fail to fit the 32k-context training on single GPU, which also requires a 16k-bps splitting for chunk-by-chunk processing. For the larger NTv2-500M, chunk size is set to 2k-bps for training. As for the t-SNE visualization of sequence-level embeddings, we deploy each pretrained baselines for extracting last layer's hidden states as the embedding for samples in training set. Embeddings of the whole training set are passed to the scikit-learn's t-SNE model (Pedregosa et al., 2011) after mean-pooling. The wall-clock time for generating all the embeddings of training set is treated as the throughput metric for each model.

**Zero-shot Regulators Classification**    The original benchmark is downloaded from here[9], we choose human's promoter dataset, multi-species promoter dataset, and multi-species epigenetic marks dataset to form the evaluation, with detailed statistical results in Table 10. The whole process

---

[8]https://github.com/HazyResearch/hyena-dna
[9]https://github.com/ChongWuLab/dna_foundation_benchmark

Table 9: Detailed statistics of vertebrate species classification

| Tasks | Sequence Length | Data Split (train/dev/test) |
|---|---|---|
| Species Classification | 1k 16k 32k | 10,000 / 1,000 / 1,000 |

for generating sequence-level embeddings is similar to vertebrate species classification task, i.e., extracting last layer's hidden states with mean-pooling for each pretrained model. The classification model is set to XGBoost. Detailed evaluation results are available in Table 15

Table 10: Detailed statistics of zero-shot regulators classification datasets

| Tasks | Sequence Length | Data Split (train/dev/test) |
|---|---|---|
| **Promoter Detection Human** | | |
| GM12878 | 102 - 2.99k (avg. 1.62k) | 10,992 / 0 / 2,750 |
| HUVEC | 101 - 2.99k (avg. 1.32k) | 11,928 / 0 / 2,982 |
| Hela-S3 | 103 - 2.99k (avg. 1.16k) | 11,736 / 0 / 2,936 |
| NHEK | 200 - 2.40k (avg. 499) | 8,170 / 0 / 2,044 |
| **Promoter Detection Multi-species** | | |
| B_amyloliquefaciens | 40 | 1,483 / 0 / 636 |
| R_capsulatus | 40 | 7,406 / 0 / 3,175 |
| Arabidopsis TATA | 251 | 3,063 / 0 / 1,313 |
| Arabidopsis no TATA | 251 | 8,267 / 0 / 3,543 |
| **Multi-species Epigenetic Marks Prediction** | | |
| 4mC_A.thaliana | | 156,697 / 0 / 67,157 |
| 4mC_C.elegans | | 84,962 / 0 / 36,398 |
| 4mC_D.melanogaster | | 126,466 / 0 / 54,200 |
| 4mC_E.coli | 41 | 8,681 / 0 / 3,721 |
| 4mC_G.pickeringii | | 24,053 / 0 / 10,309 |
| 4mC_G.subterraneus | | 63,567 / 0 / 27,243 |

**Instructed Enhancer Generation**   The whole enhancer dataset is downloaded from Huggingface Datasets[10], which includes the developmental and housekeeping activity for each enhancer fragments from Drosophila S2 cells. We maintain the original data split and fit two regression model to predict the "Dev_log2_enrichment_scaled" and "Hk_log2_enrichment_scaled", separately, with metrics set to RMSE and the Pearson correlation between the ground-truth activity values and predicted values. Details about the data split and activity distribution are available in Table 11. Since the enhancer dataset involve 400k enhancers in training set, each model is fine-tuned with 5e-5 peak learning rate, 64 batch size for 40k steps.

For the enhancer generation process, each activity value is discretized into 10 categories and directly corresponding to 10 distinct soft-prompts for instructed generation. The HyenaDNA is modified with the same soft-prompt module as HGDNA for comparison. We warp HGDNA and HyenaDNA with the GenerateMixin class from Huggingface to support cache-based beam search and sample. When generating based on the developmental activity, the beam width is set to 20 with 20 beam groups, the temperature, repetition penalty, diversity penalty, and maximum n-gram repeat length are set to 1.0, 1.2, 1.0 and 3, respectively. As for the housekeeping activity-based generation, we adjust the total beam width to 40, with other hyperparameters consistent with developmental activity-based generation. Each soft-prompt corresponds to 20 different generated enhancers, with total 200 enhancers for the final activity evaluation.

---

[10]GenerTeam/DeepSTARR-enhancer-activity

Table 11: Detailed statistics of DeepSTARR's enhancer activity dataset, with counts of training set sequences in each activity range

| Activity Ranges | Developmental | Housekeeping |
|---|---|---|
| **Sequence Length:** 249-bps | | |
| **Data Split (train/dev/test):** 402,296 / 40,570 / 41,186 | | |
| [-10, -3) | 10 | 38 |
| [-3, -2) | 5,718 | 4,080 |
| [-2, -1) | 48,862 | 39,234 |
| [-1, 0) | 160,734 | 182,936 |
| [0, 1) | 127,556 | 132,056 |
| [1, 2) | 44,080 | 23,186 |
| [2, 3) | 12,674 | 12,326 |
| [3, 4) | 2,342 | 7,728 |
| [4, 5) | 308 | 710 |
| [5, 10) | 12 | 2 |

**Ablation Studies** The major ablation studies results are included in Figure 8, which formulates a progressive workflow from the naive 30M DNABERT-2-like settings to the final HGDNA framework:

- **Tokenization and pretraining objective (stage 1)** In the first stage, it is crucial to select a proper combination of tokenizers and pretraining objectives. We set the baseline model as a 30M 6-layers full-attention (i.e., replacing all the SWA and linear attention operator to vanilla full-attention for HGDNA) with MLM objective and tokenizer from DNABERT-2. The multi-species pretraining corpus is same as HGDNA, without additional species classification auxiliary tasks and class tokens for downstream adaption, meaning that the baseline models need to follow the fine-tuning strategy from other gLM baselines, e.g., treat the start/end tokens as the [CLS] token for sequence classification. Specifically, we test the combinations of 4 tokenization strategies (BPE, non-overlapping 6-mer, base, overlapping 6-mer) and 2 pretraining objectives (MLM and CLM), these evaluations further substantiate the benefit of integrating CLM with overlapping k-mer.

- **Backbone (stage 2)** Following the results from 1st stage, the combination of overlapping 6-mer and CLM is set to the baseline for the second stage, which focuses on the backbone for token mixing. The Mamba2 and Gated DeltaNet are introduced as options for replacing full-attention layers, with sliding window attention for the remaining layers to form the hybrid manner. The evaluation results further confirm the effectiveness of brand-new linear attention mechanisms and their hybrid versions for genomic modeling.

- **Corpus (stage 3)** Coming from the backbone ablation, the backbone of baseline is now changed to the hybrid Gated DeltaNet, we further conduct the evaluation for investigating the effectiveness of gene-only corpus and multi-species training. Testing on the GUE benchmark has confirmed the power of gene-only pretraining on multi-species corpus.

- **Auxiliary tasks (stage 4)** To further strengthen the baseline model to the final HGDNA, the species classification task is concatenated with the original CLM samples, with additional class tokens to represent the species label during pretraining. When moving to the fine-tuning stage, our evaluations have demonstrated the effectiveness of reusing these class tokens for sequence-level classification, where the basic [CLS] projection (i.e., ignore the shared class tokens) fails to fully leverage the potential of auxiliary task.

Beyond the benefit for benchmark capabilities, we further investigate the potential of unified CLM training for boosting the convergence speed in fine-tuning. As illustrated in Figure 10, reusing the updated class token as downstream's target labels can speedup the fine-tuning process, especially for the initial stage. Such benefit confirms that HGDNA can learn how to aggregate the sequence-level message into the shared class token when pretraining with auxiliary tasks.

As for the generalizability of the effect bring by overlapping k-mer, we further conduct an evaluation for the HyenaDNA for measuring whether finer-grained overlapping tokenization can benefit

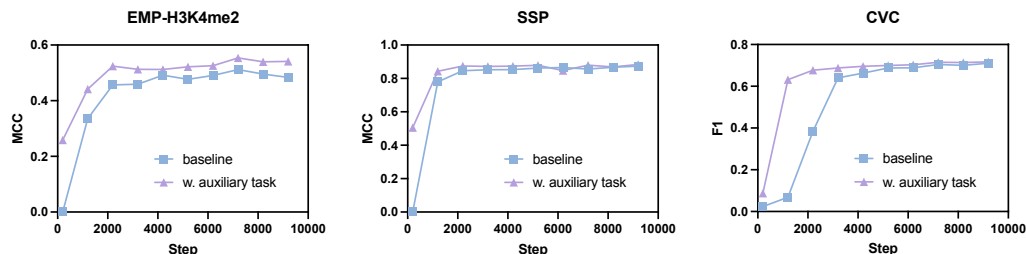

Figure 10: Effect of species classification auxiliary task and CLM-like class token reuse, it is evident that these settings jointly boost the convergence speed of HGDNA in downstream adaption

its capabilities over downstream applications. Since HyenaDNA is a CLM model that same as HGDNA, we directly replace the original base tokenizer to overlapping tokenizer for pretraining and evaluation. As the result in Table 12, models like HyenaDNA can directly benefit from overlapping tokenization by a straightforward plug-and-play manner, which proofs the superiority of overlapping tokenization compared with other methods like base, non-overlapping, and BPE.

Table 12: Effect of overlapping 6-mer tokenizer for HyenaDNA on GUE's EMP task. Both of two models are pretrained on human hg38 for 1 epoch, with training settings similar to Table 6

| HyenaDNA | H3 | H3K14ac | H3K36me3 | H3K4me1 | H3K4me2 | H3K4me3 | H3K79me3 | H3K9ac | H4 | H4ac |
|---|---|---|---|---|---|---|---|---|---|---|
| base | 75.88 (1.16) | 42.75 (0.39) | 50.73 (0.90) | 43.80 (0.39) | 30.34 (2.13) | 34.67 (2.13) | 64.18 (0.56) | 54.38 (3.39) | 79.56 (0.80) | 45.57 (3.99) |
| ovlp-6mer | **76.16 (1.05)** | **51.72 (0.22)** | **53.80 (0.72)** | **44.06 (0.36)** | **35.18 (1.02)** | **41.60 (3.74)** | **65.58 (0.83)** | **54.92 (2.83)** | **79.92 (0.44)** | **49.88 (3.55)** |

## C  PROOF OF THE SUPERIORITY OF OVERLAPPING TOKENIZATION IN MEAN-POOLED SEQUENCE SEPARABILITY

Given a sufficiently long DNA sequence $S = \{s_1, \cdots, s_i, \cdots, s_n\}$ ($s_i \in \{$A,T,C,G$\}$), herein, we consider two types of tokenization strategies:

- **Base Tokenization** $\mathcal{F}_1(\cdot)$ It provides a point-by-point mapping of the input $s_i$ to their IDs, e.g., $\mathcal{F}_1(s_i) = \mathbf{s}_i \in \{$0,1,2,3$\} \in \mathbb{Z}^{+4}$

- **Overlapping Tokenization** $\mathcal{F}_2(\cdot)$ It provides a sliding window mapping of the input sequence $S$ with window size $k \geqslant 2$. For the subsequence $S[i : i + k]$ at position $i$, it maps them to their unique IDs, i.e., $\mathcal{F}_2(S[i : i + k]) = \mathbf{s}_i \in \mathbb{Z}^{+4^k}$

It is obvious that the mapping function $\mathcal{F}_1$ can be treated as a linear transformation from $\mathcal{F}_2$, with the transformation $\mathcal{P}: \mathbb{Z}^{+4^k} \to \mathbb{Z}^{+4}$ as follows:

$$\mathcal{F}_1(S) = \mathcal{P}(\mathcal{F}_2(S))$$

where $\mathcal{F}_2$ cannot be represented as a transformation from $\mathcal{F}_1$ because the mapping space of $\mathcal{F}_2$ is larger than $\mathcal{F}_1$.

Now, consider a mean-pooling classifier $f$ in $\mathcal{F}_1(S)$ with vocabulary embedding operator $\mathbf{e}(\cdot)$:

$$f(\mathbf{e}(\mathcal{F}_1(S))) = \frac{1}{n} \sum_i (\mathbf{e}(\mathcal{F}_1(s_i))) = \frac{1}{n} \sum_i (\mathbf{e}(\mathcal{P}(\mathcal{F}_2(s_i))))$$

the separability of $\mathcal{F}_2$ is at least as good as $\mathcal{F}_1$ in linear classification models.

However, when the classification is related to ordering, $\mathcal{F}_2$ can provide a better separability than $\mathcal{F}_1$. For example, consider $S_1 = (\text{AT})^n$ and $S_2 = (\text{AATT})^{\frac{n}{2}}$ (superscript $n$ denotes n-fold repetition of the string), the $f(\mathcal{F}_1(S_1)) = \frac{1}{n} \sum \mathbf{e}([0, 1, 0, 1 \cdots, 0, 1, 0, 1])$ is equal to $f(\mathcal{F}_1(S_2)) =$

$\frac{1}{n}\sum \mathbf{e}([0,0,1,1\cdots,0,0,1,1])$, but $\mathcal{F}_2$ with $k=2$ includes the following mapping:

$$\mathcal{F}_2(\text{AT}) = 0, \mathcal{F}_2(\text{AA}) = 1, \mathcal{F}_2(\text{TT}) = 2$$

$$f(\mathbf{e}(\mathcal{F}_2(S_1))) = \frac{1}{n}\sum \mathbf{e}([0,\cdots,0])$$

$$f(\mathbf{e}(\mathcal{F}_2(S_2))) = \frac{1}{n}\sum \mathbf{e}([1,2\cdots,1,2])$$

which can distinguish between $S_1$ and $S_2$.

Overall, we can conclude that the overlapping tokenization can provide a better separability than the base tokenization in mean-pooling linear classification.

## D  ADDITIONAL CLARIFICATION

**LLM Usage**  LLMs were used throughout this work to assist in the writing and polishing process, including for both the main text and appendices. The primary approach involved using LLMs to refine our initial drafts, followed by a manual comparison of the versions to improve the clarity and quality of the writing. It is important to note that the LLM was not involved in the ideation, research methodology, or experimental design. All research concepts, ideas, and analyses were developed and conducted by the authors. The contributions of the LLM were solely focused on improving the linguistic quality of the paper, with no involvement in the scientific content or data analysis. The authors take full responsibility for the content of the manuscript, including any text generated or polished by the LLM. We have ensured that the LLM-generated text adheres to ethical guidelines and does not contribute to plagiarism or scientific misconduct.

**Ethics Statement**  This work adheres to the ICLR Code of Ethics. In this study, no human subjects or animal experimentation was involved. All datasets used were sourced in compliance with relevant usage guidelines, ensuring no violation of privacy. We have taken care to avoid any biases or discriminatory outcomes in our research process. No personally identifiable information was used, and no experiments were conducted that could raise privacy or security concerns. We are committed to maintaining transparency and integrity throughout the research process.

**Reproducibility Statement**  We have made every effort to ensure that the results presented in this paper are reproducible. The training code and Huggingface formatted model class are available in the supplementary materials. All open-sourced datasets used in this work are publicly accessible. The experimental setup, including training steps, model configurations, and hardware details, is described in detail in the paper. We believe these measures will enable other researchers to reproduce our work and further advance the field.

# E SUPPLEMENTAL RESULTS

Table 13: Detailed results for GUE benchmark

| Dataset | NTv2-100M | NTv2-500M | DNABERT-2 | GENA-LM | HyenaDNA | Caduceus-Ps | HGDNA |
|---|---|---|---|---|---|---|---|
| **EMP** | | | | | | | |
| H3 | 72.34 (2.00) | 77.82 (1.10) | 77.84 (3.74) | 75.94 (2.96) | 71.87 (0.49) | 76.59 (1.49) | **78.35 (2.17)** |
| H3K14ac | 53.34 (0.42) | 55.92 (0.30) | 52.27 (9.00) | 38.64 (1.53) | 41.59 (0.58) | 49.94 (1.13) | **70.97 (0.92)** |
| H3K36me3 | 58.81 (0.40) | 62.89 (0.05) | 54.51 (8.31) | 46.58 (1.02) | 46.24 (2.31) | 53.82 (1.57) | **69.82 (1.77)** |
| H3K4me1 | 50.93 (2.90) | 55.08 (0.20) | 48.13 (7.73) | 38.56 (1.09) | 40.17 (0.26) | 49.79 (1.72) | **57.27 (0.07)** |
| H3K4me2 | 29.41 (1.33) | 35.36 (2.03) | 39.14 (4.35) | 29.23 (0.01) | 29.27 (1.07) | 32.34 (1.24) | **55.55 (1.99)** |
| H3K4me3 | 34.33 (17.36) | 35.52 (6.02) | 37.56 (8.20) | 16.72 (3.59) | 27.74 (7.14) | 33.22 (1.96) | **63.67 (0.22)** |
| H3K79me3 | 60.36 (2.66) | 65.90 (0.42) | 63.37 (6.27) | 60.50 (0.05) | 60.69 (0.05) | 62.98 (0.76) | **71.32 (1.48)** |
| H3K9ac | 54.20 (1.04) | 57.83 (2.74) | 56.25 (2.73) | 51.32 (1.48) | 50.28 (2.45) | 56.71 (1.94) | **67.79 (1.62)** |
| H4 | 80.98 (0.85) | 80.13 (1.01) | 80.41 (0.30) | 78.41 (0.43) | 76.79 (1.72) | 81.40 (0.15) | **83.05 (0.79)** |
| H4ac | 45.60 (8.85) | 49.91 (1.87) | 50.72 (8.13) | 34.29 (0.94) | 38.60 (6.87) | 43.13 (1.31) | **66.26 (0.82)** |
| **TFM** | | | | | | | |
| 0 | 53.49 (0.84) | **62.11 (5.76)** | 56.76 (3.24) | 33.69 (8.05) | 54.34 (3.33) | 52.26 (6.63) | 62.03 (0.99) |
| 1 | 82.09 (0.40) | **85.34 (1.12)** | 85.13 (5.27) | 82.21 (0.23) | 80.82 (0.26) | 84.18 (1.58) | 85.18 (0.41) |
| 2 | 79.26 (0.63) | 82.33 (1.82) | 79.64 (9.49) | 81.71 (1.76) | 78.77 (2.94) | 82.33 (1.00) | **82.50 (1.65)** |
| 3 | 72.80 (2.78) | 80.88 (4.16) | 80.97 (5.94) | **83.67 (3.98)** | 70.77 (2.35) | 83.27 (0.82) | 83.36 (0.80) |
| 4 | 40.53 (1.98) | 47.30 (2.55) | **52.53 (0.11)** | 42.18 (3.51) | 42.66 (2.64) | 46.68 (0.29) | 48.61 (0.54) |
| **CVC** | | | | | | | |
| | 61.93 (3.64) | 63.47 (0.30) | 71.02 (1.28) | – | 46.48 (6.00) | 68.64 (3.54) | **72.50 (0.37)** |
| **TFH** | | | | | | | |
| 0 | 66.38 (1.67) | 64.46 (3.44) | **69.37 (2.60)** | 69.02 (1.02) | 63.75 (2.60) | 68.82 (0.99) | 67.84 (0.66) |
| 1 | 68.89 (1.47) | 71.49 (3.88) | 71.96 (0.36) | 71.65 (1.40) | 69.30 (0.19) | **73.37 (0.24)** | 70.50 (1.49) |
| 2 | 61.53 (4.14) | **66.99 (1.01)** | 62.66 (10.67) | 60.70 (0.51) | 64.26 (2.94) | 64.67 (0.35) | 65.60 (7.04) |
| 3 | 53.80 (8.48) | 56.47 (7.94) | **62.40 (10.66)** | 48.81 (3.17) | 31.23 (17.35) | 51.04 (1.38) | 46.17 (12.11) |
| 4 | 70.20 (2.19) | 77.40 (0.20) | **78.40 (3.57)** | 71.75 (2.00) | 71.93 (1.96) | 78.20 (3.20) | 73.80 (4.60) |
| **PD** | | | | | | | |
| all | 90.29 (0.52) | **93.17 (0.38)** | 87.57 (8.22) | 85.35 (1.07) | 83.72 (0.50) | 83.78 (0.98) | 90.68 (1.00) |
| tata | 71.64 (11.73) | **86.94 (0.22)** | 57.42 (15.47) | 64.14 (5.16) | 57.66 (3.80) | 50.31 (0.34) | 75.60 (10.85) |
| no tata | 93.85 (1.42) | **94.08 (0.56)** | 93.70 (0.63) | 93.89 (0.12) | 90.56 (0.37) | 92.88 (0.41) | 93.22 (0.34) |
| **CPD** | | | | | | | |
| all | 68.94 (0.53) | **72.01 (1.97)** | 68.52 (7.69) | 63.06 (0.39) | 66.62 (1.76) | 64.12 (1.75) | 71.44 (2.55) |
| tata | 76.52 (5.21) | **79.06 (6.60)** | 71.73 (2.30) | 70.98 (0.96) | 74.02 (0.20) | 69.52 (2.20) | 75.68 (0.76) |
| no tata | 69.32 (0.11) | **73.15 (0.07)** | 68.60 (2.39) | 67.45 (0.64) | 68.17 (0.96) | 67.52 (0.23) | 70.88 (1.60) |
| **SSP** | | | | | | | |
| | 86.00 (1.84) | **90.49 (0.89)** | 86.15 (1.86) | 76.55 (2.55) | 83.20 (5.97) | 83.51 (0.54) | 90.20 (0.88) |

Table 14: Detailed results in Nucleotide Transformer (revised) benchmark

| Dataset | NTv2-100M | NTv2-500M | DNABERT-2 | GENA-LM | HyenaDNA | Caduceus-Ps | HGDNA |
|---|---|---|---|---|---|---|---|
| **EMP** | | | | | | | |
| H2AFZ | 51.09 (0.38) | **53.44 (0.96)** | 52.53 (0.87) | 48.62 (1.84) | 49.42 (2.26) | 48.66 (4.31) | 51.18 (0.86) |
| H3K27ac | 47.39 (3.73) | 48.29 (0.11) | 52.10 (1.43) | 51.75 (1.78) | 50.49 (5.77) | 46.80 (3.95) | **53.06 (5.79)** |
| H3K27me3 | 59.12 (1.12) | **61.53 (0.49)** | 59.48 (1.15) | 59.91 (0.19) | 57.38 (0.06) | 55.69 (0.69) | 59.25 (0.56) |
| H3K36me3 | 60.94 (1.37) | **64.39 (1.60)** | 59.98 (3.80) | 61.73 (1.01) | 54.31 (3.26) | 56.36 (2.20) | 58.54 (1.28) |
| H3K4me1 | 45.71 (3.90) | **50.83 (0.47)** | 49.68 (0.85) | 47.18 (1.34) | 43.43 (1.94) | 42.21 (0.36) | 50.12 (2.11) |
| H3K4me2 | 54.30 (1.63) | 57.77 (2.91) | 58.06 (4.21) | 55.09 (0.98) | 52.68 (0.41) | 47.64 (2.47) | **58.57 (1.28)** |
| H3K4me3 | 62.11 (0.52) | 65.24 (0.55) | 64.97 (3.59) | **66.28 (0.80)** | 59.79 (2.32) | 62.42 (0.82) | 64.19 (0.32) |
| H3K9ac | 54.79 (2.30) | 56.67 (1.49) | 55.25 (0.54) | **56.99 (2.13)** | 52.19 (1.92) | 50.71 (4.00) | 55.23 (0.98) |
| H3K9me3 | 45.70 (3.11) | 45.88 (3.29) | **50.25 (1.04)** | 47.41 (0.39) | 43.27 (3.61) | 44.47 (1.62) | 43.45 (2.48) |
| H4K20me1 | 64.66 (1.98) | 63.42 (3.56) | **65.94 (0.47)** | 65.15 (2.15) | 63.37 (1.23) | 61.74 (2.92) | 62.74 (0.76) |
| **ED** | | | | | | | |
| Enhancer | 58.17 (0.33) | **59.04 (2.35)** | 54.43 (2.85) | 53.32 (1.84) | 52.29 (0.69) | 48.66 (1.09) | 55.12 (2.19) |
| Enhancer type | 53.56 (0.70) | **54.02 (1.39)** | 50.39 (1.35) | 50.25 (1.01) | 48.96 (0.54) | 46.48 (0.66) | 52.16 (2.10) |
| **PD** | | | | | | | |
| all | 77.81 (0.47) | **79.17 (1.95)** | 75.42 (0.34) | 71.21 (2.54) | 75.26 (2.15) | 72.07 (0.13) | 73.21 (0.58) |
| tata | 92.45 (3.78) | 92.51 (2.81) | 83.03 (4.70) | 84.90 (0.95) | 87.92 (2.79) | 78.47 (2.15) | **93.40 (3.68)** |
| no tata | 76.19 (0.82) | **80.36 (0.19)** | 75.11 (0.29) | 73.64 (1.13) | 73.67 (0.74) | 75.12 (0.40) | 74.71 (2.21) |
| **SSP** | | | | | | | |
| all | 95.99 (0.86) | **97.20 (0.36)** | 83.98 (3.76) | 77.17 (1.15) | 82.50 (2.64) | 80.98 (1.26) | 96.19 (0.31) |
| acceptor | 95.87 (0.13) | **96.53 (0.26)** | 82.57 (1.43) | 81.34 (2.01) | 75.17 (3.28) | 78.72 (2.54) | 92.74 (3.33) |
| donor | 97.20 (0.20) | **97.46 (0.27)** | 83.40 (3.73) | 84.08 (2.56) | 89.06 (4.28) | 76.19 (5.40) | 96.40 (0.07) |

Table 15: Detailed results in zero-shot regulators classification benchmark. There are no significant standard deviations in this benchmark

| Dataset | NTv2-100M | NTv2-500M | DNABERT-2 | HyenaDNA | Caduceus-Ps | HGDNA |
|---|---|---|---|---|---|---|
| **PDH** | | | | | | |
| GM12878 | 86.61 | **88.22** | 85.96 | 86.11 | 88.07 | 86.55 |
| HUVEC | 89.40 | **90.61** | 89.43 | 89.21 | 89.74 | 90.27 |
| Hela-S3 | 87.09 | 87.75 | 87.07 | 87.83 | **89.45** | 88.71 |
| NHEK | 71.36 | 72.12 | 73.41 | 73.05 | **77.89** | 76.38 |
| **PDM** | | | | | | |
| B_amyloliquefaciens | 45.00 | 50.00 | 45.05 | 58.49 | 58.17 | **60.10** |
| R_capsulatus | 21.09 | 21.49 | 18.58 | 22.50 | 24.41 | **33.65** |
| Arabidopsis tata | 76.82 | 77.55 | 69.70 | 73.94 | 78.13 | **81.94** |
| Arabidopsis no tata | 78.90 | 78.04 | 72.45 | 76.19 | **82.40** | 81.30 |
| **EMP** | | | | | | |
| 4mC_A.thaliana | 12.22 | 14.36 | 11.12 | 3.79 | 3.26 | **16.98** |
| 4mC_C.elegans | 14.34 | 17.65 | 11.33 | 7.75 | 7.53 | **18.09** |
| 4mC_D.melanogaster | 16.11 | 19.45 | 13.87 | 3.28 | 3.41 | **21.13** |
| 4mC_E.coli | 6.21 | 2.42 | 5.78 | 0.06 | 0.81 | **9.39** |
| 4mC_G.pickeringii | 8.37 | **9.45** | 5.93 | 5.24 | 5.13 | 9.25 |
| 4mC_G.subterraneus | 4.60 | 6.39 | 4.37 | 1.41 | 1.39 | **8.48** |

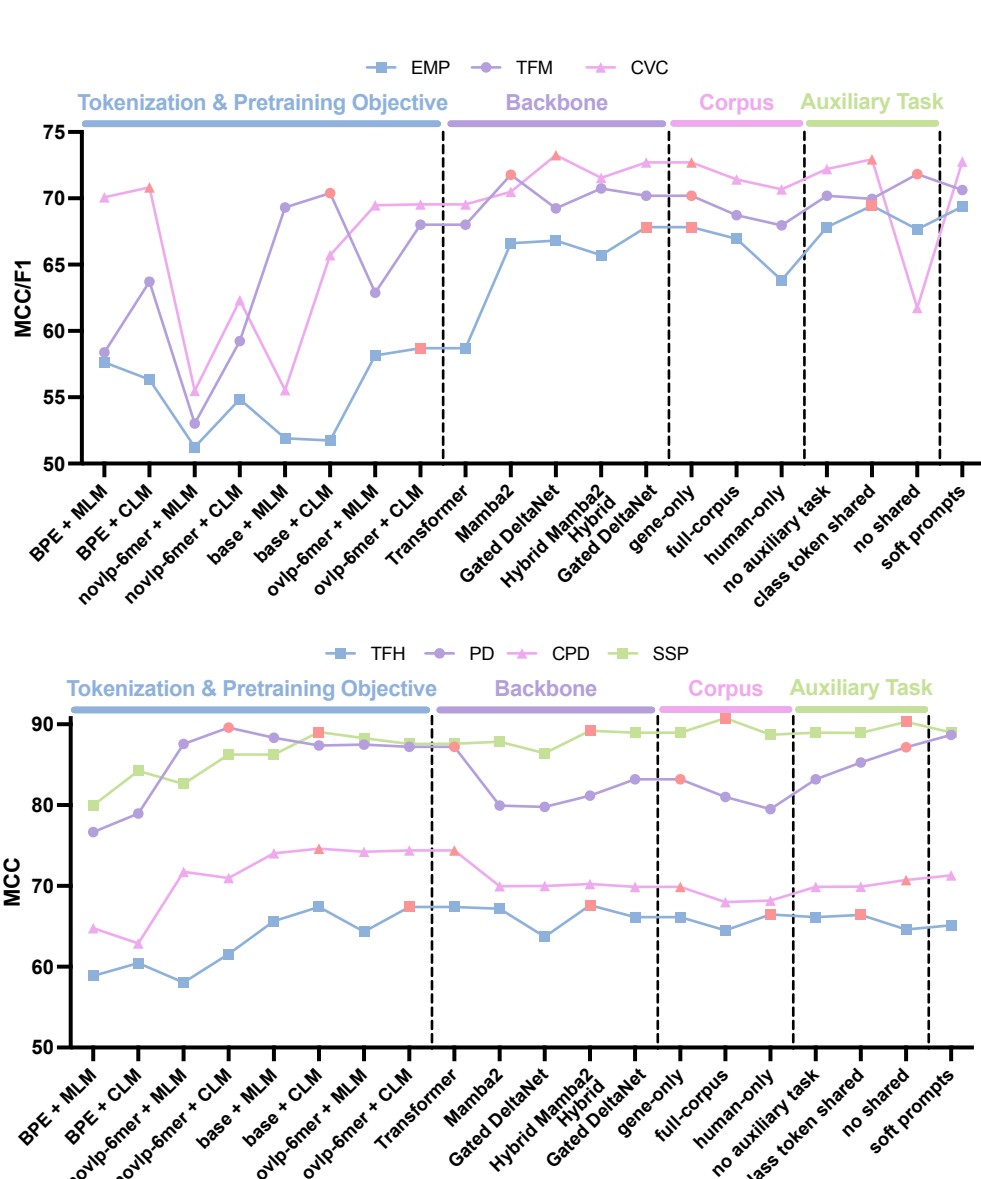

Figure 11: Detailed GUE Benchmark ablation results for Figure 8, with upper figure for 3 multi-species task and lower figure for 4 human-oriented task. Each task's best settings in different stage are highlighted in red

