# OpenReview forum: "Beyond the Bases: Unleashing Overlapping DNA Tokenization via Unified Linear-Time Autoregressive"
_ICLR.cc/2026/Conference — ICLR 2026 Conference Withdrawn Submission_

### Official Review · Reviewer_QiW9 · 2025-10-24

**Soundness:** 3
**Presentation:** 3
**Contribution:** 2
**Rating:** 4
**Confidence:** 4

**Summary:**

This paper revisits overlapping k-mer tokenization in genomic language models and proposes a unified autoregressive framework combined with a hybrid backbone that uses SWA with Gated DeltaNet. The authors argue that MLM objectives under overlapping tokenization cause information leakage, while CLM naturally avoids this issue and enables consistent pretraining–finetuning alignment through shared class tokens. The model achieves competitive results on the GUE and NT benchmarks, plus long-context benchmarks and enhancer generation using DeepSTARR. Despite its small size and low cost, HGDNA outperforms or matches much larger BERT-like models.

**Strengths:**

1. The paper analyzes the well-known information leakage problem in overlapping k-mer tokenization for MLM-style pretraining and shows that CLM provides a principled fix. This revisit is meaningful and provide a replacement to the current trending tokenization methods.

2. With only 30M parameters, the model achieves results comparable to 100M–500M Transformer baselines. Linear attention and gated-delta updates yield substantial throughput gains.

3. The authors include short-range (GUE/NT) and long-range (Gener, vertebrate classification) evaluations, as well as enhancer generation tasks, comprehensively evaluated a gLM in multiple perspectives.

**Weaknesses:**

1. The long-context tasks test sequence length handling, not regulatory or 3D genomic reasoning. Thus, “long-range” here reflects context size, not biological distance reasoning. As the model is only trained on gene region, the gene-related tasks could have higher performance, but many distal regulation related tasks could be affected.

2.As the major contribution is tokenization ,the paper claims advantages for overlapping k-mer tokenization, but does not compare against recent alternatives such as MxDNA, where some adaptive tokenization method is used to learn the best tokenizer.

**Questions:**

1. Can the author provides some demonstration on long-range regulatory benchmarks like in Genomics LRB or DNALongBench (e.g. eQTL tasks) to demonstrate that gene-oriented training would also give competitive performance on other kinds of long-range tasks?

2. Can the author compare with some newer tokenization methods like MxDNA to see the difference between overlapping k-mer and some new adaptive tokenization methods?

---

### Official Review · Reviewer_UGp7 · 2025-10-28

**Soundness:** 2
**Presentation:** 3
**Contribution:** 3
**Rating:** 6
**Confidence:** 4

**Summary:**

This submission presents HGDNA, a 30M genomic language model with overlapping k-mer tokenization and sliding window attention-based Gated DeltaNet backbone. It identifies that overlapping k-mer has been largely neglected due to its incompatibility with standard Masked Language Modeling objectives, which suffer from information leakage. To address this, HGDNA is pretrained on a large 13.6B-bps gene-annotated corpus and fine-tuned in a unified causal language modeling (CLM) paradigm, with auxiliary species classification and shared classification tokens. Experiments show competitive performance on both short- and long-range genomic benchmarks, zero-shot embedding and sequence generation tasks. It offers improvements in efficiency and downstream performance compared to existing BERT-like and linear-attention genomic language models, and provides mathematical and empirical justification for the overlapping k-mer tokenization.

**Strengths:**

**(S1)** This work clearly identifies the under-utilization of overlapping k-mer tokenization in genomic language models. It offers a thorough re-examination in Sec. 3.1, providing both theoretical proof in Appendix C and empirical evidence in Fig. 4 for overlapping k-mer’s benefits in sequence separability and classification.

**(S2)** The integration of species classification as an auxiliary signal in a unified CLM framework is well-motivated and empirically validated in ablations (Fig. 8). It is a simple, effective solution to a known problem and appears to bridge pre-training and downstream adaption and accelerates fine-tuning as shown in Fig. 10.

**(S3)** Competitive results with impressive parameter efficiency. HGDNA delivers competitive results with only 30M parameters, notably outperforming HyenaDNA and Caduceus-Ps (parameter-efficient models) and surpassing BERT-like models that use 10–50x more training tokens. Fig 6 uses t-SNE to illustrate the improved species separability of the learned representations. The progressive ablation study in Fig. 8 is also excellent.

**(S4)** Clear method formulation. The core equations for both attention (Sec. 2) and Gated DeltaNet memory update (including the role of $\alpha_t$, $\beta_t$; Appendix A) are well-presented.

**Weaknesses:**

**(W1)** Incomplete literature review. The manuscript does not discuss recent studies on different genomic tokenization and overlapping k-mers [1] [2] [3] [4] [5] [6]. Many of these go into detail about overlapping k-mer tradeoffs or explores different learnable tokenization methods. I encourage the authors to include related discussions in the revised manuscript for better literature review.

**(W2)** Definition in overlapping k-mer CLM. Sec. 3.1, Appendix A, and Fig. 2 show the performance and rationale for switching from MLM to CLM using overlapping k-mers. But there is ambiguity in the exact token prediction target space transition. How is the prediction target precisely mapped when transitioning from a large $4^k$ vocabulary to single-nucleotide space? The loss surface discontinuity and “sollapse” mentioned are shown empirically, but the theoretical part for such behavior in the context of embedding smoothness or probability simplex transitions is not discussed. Furthermore, negative sampling and class imbalance handling in $\mathbb{V}^c$ are only briefly described; more clarity on their implementation and theoretical justification is needed.

**(W3)** Fig. 8 and Tab. 12 show that plugging overlapping k-mer tokenization into HyenaDNA yields performance gains. However, there is no comparison against other overlapping k-mer strategies in open literature or across diverse model architectures (e.g., ProkBERT, Pangenome LMs). This limits the ability to separate the true gains vs. combination effects with the hybrid attention backbone.

**(W4)** Generalization to longer sequences. The maximum pretraining context is 2,048-bps, whereas evaluation (e.g., in the Gener or vertebrate species tasks) extends to 32k/96k. IMHO, an analysis of performance degradation, memory scaling, and the representations' extrapolation robustness when reusing relatively short-context-pretrained models for long sequences would help support the claims.

**(W5)** In Sec. 3.2 and Appendix A, there is some ambiguity in the precise indexing schemes and batching for class tokens and soft prompts, especially how $\mathbb{V}^c$ is actually shared or (re-)initialized at pretraining/fine-tuning boundaries. The approach relies heavily on mean-pooling for sequence representations, which has limitations unaddressed in discussions; the impact of alternative pooling or attention-based aggregation is not explored.

**(W6)** High-moment or distributional analysis. Most of the evaluations focus on mean performance (MCC/F1). There is not much analysis of tails (e.g., rare regulatory elements, low-resource species, or out-of-distribution chromosomes). The applicability for “hard” cases is claimed, but rarely quantified.


---
## Reference
[1] Impact of Tokenizer Selection in Genomic Language Models, Bioinformatics 2025. It should be included in the tokenization discussion in Sec. 2 for overlapping k-mers vs other strategies.

[2] Tokenization and Deep Learning Architectures in Genomics: A Comprehensive Review, Computational and Structural Biotechnology Journal, 2025. It discusses the tradeoffs in overlapping tokenization.

[3] Genomic Language Models: Opportunities and Challenges, Trends in Genetics 2025. It addresses broader architecture and tokenization considerations for genomic language models.

[4] VQDNA: Unleashing the Power of Vector Quantization for Multi-Species Genomic Sequence Modeling, ICML 2024. It explores learnable tokenization vocabularies in genomic language models which shows a path beyond fixed tokenizers like BPE and k-mer.

[5] Model Decides How to Tokenize: Adaptive DNA Sequence Tokenization with MxDNA, NeurIPS 2024. It aims to address the limitations of NLP-derived tokenization methods by explicitly modeling discontinuous, overlapping, and ambiguous biological units

**Questions:**

Most of my concerns and related suggestions have been stated in the Weaknesses section. I encourage the authors to focus their efforts on addressing those points, as they are critical for strengthening the manuscript in the rebuttal stage. The following are more specific, minor questions to help the authors think more deeply about certain design choices and experiment setups, which I hope could be helpful for this and future work:

**(Q1)** Can the authors formalize the predicted outputs’ transition between the large $4^k$-sized vocabulary and the single nucleotide space in the CLM scenario? Is there an underlying theoretical result supporting the smoothness or stability of this transition, or an explanation for the observed step in loss convergence?

**(Q2)** What is the empirical/quantitative deterioration (if any) in model performance as the deployed sequence length increases beyond the pretraining maximum? Can the authors provide explicit analysis (Table or plot) of performance drop or extrapolation robustness at 32k/96k context?

**(Q3)** Do the authors anticipate further gains from exploring more complex sequence aggregation methods than mean pooling, given the known weaknesses of mean pooling in representations for variable-length genomic signals?

**(Q4)** Can the authors provide the precise initialization, interaction, and training regime for the shared class vocabulary $\mathbb{V}^c$ across pretraining and downstream tasks? How does it compare to class token approaches explored in protein modeling or transcriptomics?

---
## Justification:

I first give a rating of 6, primarily due to the clear motivation, the reasonable choice of hybrid architecture, and the impressive results and parameter efficiency. In particular, the performance suggests the potential practical value, which is critical for AI4Science and genomics. I would be glad to raise my rating if thoughtful responses and improvements are provided. Conversely, if most of the concerns remain unaddressed, I may consider lowering my score. I am also open to follow-up discussions with the authors to help further strengthen this work.

I hope these comments help my fellow reviewers and ACs understand the basis of my recommendation.

---

### Official Review · Reviewer_xSWf · 2025-11-01

**Soundness:** 3
**Presentation:** 2
**Contribution:** 2
**Rating:** 4
**Confidence:** 3

**Summary:**

This paper introduces HGDNA, a hybrid genomic language model that leverages the use of overlapping k-mers with the Gated DeltaNet model. under a unified causal language modeling framework.

**Strengths:**

- HGDNA appears to be competitive with much larger genomic language models on a series off classification and zero-shot embedding tasks with much lower training budgets.
- The authors provide a comprehensive set of experiments and ablations to evaluate the model.

**Weaknesses:**

- While the paper presents the use of CLM on genome sequences as a contribution, this has already been demonstrated by the HyenaDNA paper.
- In the NT and GUE benchmarks, HGDNA shows limited or inconsistent advantages on shorter sequences, which weakens the argument for its general superiority.
- Ultimately, the contributions of the paper seem very incremental and there doesn’t seem to be a clear advantage to using HGDNA. The paper would be strengthened if the authors further investigate the use cases where the model is particularly strong.

**Questions:**

- Are the GUE Benchmark and NT benchmarks overlapping? Please confirm that the data used for the benchmarks is non-overlapping, to avoid redundancies in performance reporting.
- How does HGDNA compare when trained or evaluated on the full dataset scale used by larger models?
- Which aspects of HGDNA’s design are most responsible for its improvements in zero-shot and long-range tasks? Providing biological or interpretive insights would be helpful.
- How is the efficiency reported in Figure 5 computed? Please include quantitative runtime and memory comparisons against DNABERT-2 and HyenaDNA to validate the claimed 25× speedup.

---

### Official Review · Reviewer_23qh · 2025-11-06

**Soundness:** 3
**Presentation:** 1
**Contribution:** 3
**Rating:** 2
**Confidence:** 4

**Summary:**

The paper presents a model which using state-space (linear-attention) in combination with sliding window attention (Gated DeltaNet), trained on DNA data.

**Strengths:**

The paper presents a performant model, using a state-of-the-art state-space/linear-attention architecture, trained and evaluated on challenging DNA dataset.

The experiments include baselines on comparable and competitive state-of-the-art models.

**Weaknesses:**

**Novelty**
- The authors describe their model as being Gated DeltaNet plus sliding window attention (L178, L195). But the Gated DeltaNet paper already includes sliding window attention. As far as I can tell, there is no methodological novelty in the architecture, but the paper does not make this clear or easy to assess.
- The other methodological novelty which the authors appear to put forward is their masking strategy, which is a blocking mask instead of uniform random mask. This is obviously necessary when using overlapping k-mers. It was not really necessary to do any experiments on a masking strategy where 100% of the information in the masked out tokens is leaked to the model via the two neighbouring tokens in order to conclude this was a poor pretraining method. You do not need to run an experiment for which the answer is known to all before it is run. The authors describe their masking method as a "causal" mask (L019), but train a bi-directional model, which is a contradiction. Based on Fig 2, the masking strategy appears to simply be a blocking mask, with its length informed by the amount of overlap of the overlapping tokenization. Note that blocking masks have been done previously in the literature, in various domains e.g. by SpanBERT (Joshi et al, 2019).
- The masking strategy should be compared with the RandomMask strategy (Liang et al, 2023), which was proposed explicitly for changing the size of blocking masks while training DNA models.

**Soundness**
- Fig 1 shows downstream performance vs number of training tokens and model size. However, the method presented by the authors uses overlapping k-mer tokens, so it has more input tokens than models that use non-overlapping k-mer tokens. This means that it may use more compute per sample seen, therefore it would be advantageous to also show a similar graph with the amount of training FLOPs on the x-axis. This would be advantageous to the authors since their method requires fewer FLOPs to train despite the overlap in tokens and thus increased token sequence length.

**Presentation**
- The use of a bespoke name for the model (HGDNA) which is unrelated to the existing architecture which is being deployed and trained (Gated DeltaNet) obfuscates the fact that the architecture is functionally unchanged from Gated DeltaNet.
- There are a lot of grammatical errors in the paper. I think the average number of errors per paragraph is higher than one. Egregious examples include the background section, where L107-116 contains around 8 grammatical errors (an average of one per line). Better care needs to be taken in the writing. The authors might want to ask a native speaker to proof-read the paper, or use a tool such as Grammarly to help identify and fix the issues. Unfortunately they are too numerous for me to list here.
- Even the title of the paper contains a grammatical error. "Autoregressive" is an adjective, not a noun and so it needs to be followed by a noun, i.e. "Unleashing Overlapping DNA Tokenization via Unified Linear-Time Autoregressive [Modelling]", or rephrased to use a noun instead of an adjective i.e. "Unleashing Overlapping DNA Tokenization via Unified Linear-Time Autoregression".
- This is a subtle difference, so I will flag it explicitly (most grammatical errors in the paper are quite obvious and not subtle). Section heading "Data-dependency Memory Management" (L119) should be "Data-dependent Memory Management" instead. Both are legitimate phrases, but they mean different things. "Data-dependency Memory Management" means that memory is managed in a way to facilitate data dependencies, i.e. moving memory around in RAM so that data order relationships can be better facilitated. "Data-dependent Memory Management" means the management process for the memory depends on the data.
- Fig 8. Unclear what is being shown. Why is it a line plot? The x-axis contents are discrete - not only is the domain not continuous, it is not ordinal either, so how are the discrete values ordered and why can you join them up with a linearly interpolating line? The use of a line makes it look like you traverse the ablation from left to right, but the labels do not agree with this. There is insufficient detail provided to infer whether ablations are applied independently are accumulatively.

*Minor*
- Fig 4b
    - Label "base" is unclear. (This could mean nucleotide or baseline)
    - This is not really the right type of plot. The value is a ratio and can be any value from 0 to +infty. The value of 0 is just as impossible to achieve as +infty, so this shouldn't be a bar chart with bars going up from 0. The plot should arguably show the log of the ratio instead, since that measurement is symmetric - 1.1 is comparable with 1/1.1, but 0.909 and 1.1 are different distances away from 1 whereas log(1.1) = -log(1/1.1) which is symmetric. If the plot is not changed to show the log-ratio, at the least the bars should go up/down from the expected value of 1.0 instead of up from 0. This makes it clearer that negative values are worse than chance level for separability.
    - The plot would be further improved if it had error bars to indicate the variability across classes.
- Fig 4c colours should be consistent with Fig 4b (don't re-use pink to plot a different method within the same figure)
- Fig 6. Why have you picked these colours? The blue, cyan, and green are quite easily confused and unclear. A different palette would be better.
- L234 (etc) quotation marks are the wrong way around
- L275 This notation abuses the less than and greater than signs. It would be more appropriate to use langle and rangle here, i.e. $\langle \text{start token} \rangle \langle \text{gene1} \rangle | \langle \text{gene2} \rangle \cdots$.
- URLs in the reference section are not clickable links, hindering their utility. It would be better if DOIs were clickable too, which can be achieved just by importing the doi package
- Please check the formatting of citations, e.g. L658 "Ł ukasz"

**References**
- Joshi et al (2019). "SpanBERT: Improving Pre-training by Representing and Predicting Spans". ACL 2020. https://aclanthology.org/2020.tacl-1.5.pdf
- [RandomMask](https://arxiv.org/abs/2310.07644): Liang et al (2023). "Toward Understanding BERT-Like Pre-Training for DNA Foundation Models"

**Questions:**

- L067 "while contiguous masking can inevitably lead to an imbalanced masking recovery difficulty." This sentence is unintelligible. What is it supposed to say and mean?
- Fig 4a: What is the difference between the curves "MLM" and "MLM leakage"
- L255 When you say "using a randomly initialized linear layer", do you mean you used a random map of the features for XGBoost, or do you mean you trained a linear probe on top of a frozen encoder? If it is the former, I do not understand the motivation for this; if it is the latter, you should describe it as a "linear probe" or similar clearly in the text.

---

### Author Response · Authors · 2025-11-21
**Thanks for all reviewers' comments**

After careful consideration, we have decided to withdraw our submission. We sincerely thank all reviewers for the time and effort they have devoted to reviewing our paper. Their insightful, valuable, and detailed comments are greatly appreciated and will be very helpful for further improvement

Finally, in response to some of the comments from Reviewer **#23qh**, we provide the following clarifications:

1. In Figure 2, both masking strategies in the MLM part were introduced in the original DNABERT paper, with the block masking being used in the final pretraining. However, HGDNA/Gated DeltaNet is inherently an autoregressive model that solely relies on the standard causal mask, without any additional masking strategies from MLM. The bidirectional sliding window for SWA is applied only in downstream sentence-level tasks to improve performance. Moreover, as overlapping tokenization is rarely adopted in existing DNA models, the pretraining results corresponding to the vanilla MLM masking are not available at this time, which is the reason that we introduce the corresponding loss curve in Figure 4

2. The training budget is solely based on the total number of tokens fed into the DNA LM, but not the original base-pair counts. Hence, the statement "so it has more input tokens than models that use non-overlapping k-mer tokens" is not applicable to the pretraining stage

---

### Note · Authors · 2025-11-21

I have read and agree with the venue's withdrawal policy on behalf of myself and my co-authors.